# Structural basis of peptide secretion for Quorum sensing by ComA

Lin Yu[1,2,8], Xin Xu[1,8], Wan-Zhen Chua [3,8], Hao Feng[1,8], Zheng Ser[4], Kai Shao[1], Jian Shi[1,5], Yumei Wang[6], Zongli Li [7], Radoslaw M. Sobota [4], Lok-To Sham [3] ✉ & Min Luo [1,5] ✉

Quorum sensing (QS) is a crucial regulatory mechanism controlling bacterial signalling and holds promise for novel therapies against antimicrobial resistance. In Gram-positive bacteria, such as *Streptococcus pneumoniae*, ComA is a conserved efflux pump responsible for the maturation and secretion of peptide signals, including the competence-stimulating peptide (CSP), yet its structure and function remain unclear. Here, we functionally characterize ComA as an ABC transporter with high ATP affinity and determined its cryo-EM structures in the presence or absence of CSP or nucleotides. Our findings reveal a network of strong electrostatic interactions unique to ComA at the intracellular gate, a putative binding pocket for two CSP molecules, and negatively charged residues facilitating CSP translocation. Mutations of these residues affect ComA's peptidase activity in-vitro and prevent CSP export in-vivo. We demonstrate that ATP-Mg$^{2+}$ triggers the outward-facing conformation of ComA for CSP release, rather than ATP alone. Our study provides molecular insights into the QS signal peptide secretion, highlighting potential targets for QS-targeting drugs.

Triggered by small signaling molecules at a high cell density, quorum sensing (QS) allows pathogens to synchronize gene expression in response to the ever-changing environment[1–3]. Processes regulated by QS include antibiotic production[4–6], the CRISPR system[7], biofilm formation[8–10], sporulation[11,12], and competence[13–16]. Unlike QS signaling molecules in Gram-negative bacteria that can freely diffuse across the cell membrane, an active transport mechanism is required to export peptide signals in the Gram-positive QS systems[4]. For example, in the competence pathway in *S. pneumoniae*, the QS signal competence stimulating peptide (CSP) is processed and secreted by a bi-functional efflux pump ComA (Fig. 1a)[17–19]. ComA first cleaves the N-terminal leader sequence of ComC to produce CSP. The mature CSP is then actively secreted through the transmembrane conduit of ComA. Once achieved a threshold concentration, CSP activates the ComDE two-component system, leading to the expression of early competence genes. Among them are the alternative sigma factors ComX1 and ComX2. They stimulate the transcription of late competence genes including the competence pili, single-stranded DNA binding proteins, and the DNA uptake channel[20,21]. Competence is then turned off by the recombination mediator protein DprA and the cell enters a refractory period for approximately one hour[22]. Natural genetic transformation is one of the three mechanisms of horizontal gene transfer. It results in

[1]Department of Biological Sciences, Faculty of Science, National University of Singapore, Singapore 117543, Singapore. [2]Institute of Translational Medicine, Medical College, Yangzhou University, Yangzhou 225001 Jiangsu, China. [3]Infectious Diseases Translational Research Programme and Department of Microbiology and Immunology, Yong Loo Lin School of Medicine, National University of Singapore, Singapore 117545, Singapore. [4]Functional Proteomics Laboratory, SingMass National Laboratory, Institute of Molecular and Cell Biology, Agency for Science, Technology and Research (A*STAR), Singapore 138673, Singapore. [5]Center for Bioimaging Sciences, Department of Biological Sciences, National University of Singapore, Singapore 117543, Singapore. [6]Beijing National Laboratory for Condensed Matter Physics, Institute of Physics, Chinese Academy of Science, Beijing 100190, China. [7]Harvard Cryo-EM Center for Structural Biology, Harvard Medical School, Boston, MA 02115, USA. [8]These authors contributed equally: Lin Yu, Xin Xu, Wan-Zhen Chua, Hao Feng. ✉e-mail: lsham@nus.edu.sg; dbslmin@nus.edu.sg

the exchange of antibiotic-resistance genes, toxins, and virulence factors[23].

The discovery of quorum sensing (QS) in bacteria has led to the development of new strategies for combating bacterial infections[6,24,25]. One potential approach is to intercept QS, which can inhibit virulence and reduce the severity of infections. Among the early Gram-positive QS systems targeted is the accessory gene regulator (agr) QS system in *Staphylococcus aureus*, which led to the development of autoinducing peptide-based agr QS inhibitors[25–27]. Another system being explored for its therapeutic potential is the fsr QS circuit in *Enterococcus faecalis*[26,27]. Additionally, efforts are underway to attenuate virulence in pneumococcal infections[24]. However, current QS modulator development mainly focuses on disrupting the interaction between signal peptides and their receptors, rather than targeting the upstream biosynthesis process, due to the lack of underpinning molecular structures. ComA, a highly conserved membrane transporter in Gram-positive bacteria, represents an ideal drug target since signal peptide secretion is ubiquitous.

ComA is a member of the widespread PCAT (peptidase-containing ATP-binding cassette transporter) family. It consists of three domains: an N-terminal C39 cysteine peptidase domain (PEP), a C-terminal nucleotide-binding domain (NBD, or ATPase), and a transmembrane domain (TMD). The PCAT family is known for its role in peptide processing, which has been extensively studied in PCAT1[28–30]. During this process, the immature peptide binds to the cytosolic protease domain with the leader peptide and the core peptide located is positioned in the central cavity of the TMD. The mature peptide is generated by cutting off the leader peptide with the peptidase domain, followed by secretion across the membrane. The fact that all PCATs have the same architecture with a conserved peptidase domain suggests that the mechanism of peptide processing is conserved[29]. However, the process of peptide secretion may differ due to variations in the shape and size of the substrates, and this process is not yet fully understood because no structure has been obtained with the matured peptide binding.

Despite the availability of partial structures of the ComA PEP and NBD domains over a decade of studies[31–33], elucidating the mechanism of CSP binding and secretion without a full-length transporter structure is challenging, thereby limiting the development of drugs targeting the QS signal peptide secretion process. In this study, we reconstitute full-length ComA, characterize it biochemically, and solved its structure by cryo-EM. ComA is unique because it has a high affinity towards ATP. Additionally, we determined multiple structures of ComA in various conformational states, with and without CSP and nucleotides, covering critical stages of CSP export. We identify a putative noncanonical peptide binding site at the outer leaflet of the membrane. Critical residues in ComA were also validated by in-vitro peptidase assays and in-vivo CSP secretion assays in *S. pneumoniae*. Additionally, we report ComA activities were inhibited by $Zn^{2+}$, which trapped the molecule in the outward-facing conformation. ATP-$Mg^{2+}$, but not ATP, triggers the outward-facing conformation of ComA and facilitates CSP release. Our work provides a comprehensive mechanistic insight underlying the export of peptide signals for quorum sensing in gram-positive bacteria.

## Results

### Reconstitution and functional characterization of ComA
Full-length ComA, the truncated PEP domain of ComA, and its substrate ComC were overexpressed and purified by affinity and size-exclusion chromatography (Supplementary Fig. 1a). Purified ComA was solubilized in Lauryl Maltose Neopentyl Glycol (LMNG) and reconstituted in peptidiscs or nanodiscs[34]. We did not detect a significant difference in ATPase activity between LMNG solubilized ComA and ComA reconstituted in peptidiscs or nanodiscs (Fig. 1b). Moreover, the introduction of either the substrate ComC or the mature substrate CSP did not stimulate ATPase activity, a finding that aligns

with previously observed results in the study of PCAT1[29]. To show that the PEP domain remained active, the peptidase activity of ComA reconstituted in LMNG was determined by incubating the enzyme with purified ComC at different temperatures and the cleavage product was analyzed using Tris-tricine SDS-PAGE. Surprisingly, raising the reaction temperature only increased the ATPase activity of ComA but not its peptidase activity (Supplementary Fig. 1b and 1c). Specifically, the ATPase activity of ComA at 50 °C was about fivefold more than the activity at 25 °C. ComA exhibits remarkable thermostability because the ATPase activity was robust at 50 °C.

Next, the Michaelis Menten constant (Km) and the turnover rate of ComA were determined at 25 °C and 37 °C (Fig. 1c). The ATPase activity of ComA at both temperatures was fitted to the Michaelis Menten curve. Compared to most ABC transporters reported in the literature[35,36], the turnover rate of ComA for ATP was relatively low, but the affinity to ATP was higher (Km <100 μM). The peptidase activity of full-length ComA was also notably higher than the truncated peptidase domain (Fig. 1d). We also validated the two catalytic residues, Cys17 and His96, in CSP processing[31]. Indeed, changing them to Ala (C17A and H96A) abolished the ComA peptidase activity, confirming their role in substrate cleavage (Fig. 1e). In summary, our study shows that ComA, like other PCAT family members (such as PCAT1)[28,32], contains catalytic residues C17 and H96 in the peptidase domain. ATP binding inhibits its peptidase activity. Furthermore, our biochemical characterization of ComA shows that it is a unique ABC transporter with high ATP affinity and thermostability. The robustness of ComA may contribute to quorum sensing in harsh environments.

### Dissociation of the PEP domain upon ATP binding
We further investigated the coordination between the catalytic activities of the PEP and NBD domains of ComA. First, the ATPase activity was measured in the presence and absence of the substrate peptide. Neither ComC nor matured CSP affected the ATPase activity of ComA (Fig. 1b). On the other hand, adding ATP markedly inhibited the peptidase activity (Fig. 1f and Supplementary Fig. 1d). We postulated that ATP binding, likely reduces ComC cleavage. First, adding non-hydrolyzable analogs of ATP (i.e. AMP-PNP, ATPγS, and ATP-vanadate) could also inhibit the peptidase activity (Fig. 1f). The inhibition was insensitive to EDTA, which presumably inhibited the ATPase activity. Furthermore, the ComA(E647Q) mutant could also inhibit the peptidase domain to a similar extent, although this variant could only bind but not hydrolyze ATP (Supplementary Fig. 1e).

Surprisingly, we observed an inhibitory effect on the peptidase activity of the wild-type (WT) ComA enzyme in the presence of ATP-$Mg^{2+}$, compared to the enzyme in the presence of ComC alone (Fig. 1f and Supplementary Fig. 1d). We suggest that this could be attributed to ATP hydrolysis being the rate-limiting step of ComA, while the peptidase-catalyzed reaction occurs much faster. To substantiate this, we created a D646N mutant of ComA, an ATPase defective mutant that is incapable of ATP binding[37]. As anticipated, this mutant, whose ATP binding/hydrolysis activity has been abolished (Supplementary Fig. 1e), showed a restoration of peptidase activity to a level akin to the WT enzyme in the presence of ComC alone (Fig. 1f and Supplementary Fig. 1d). This supports our hypothesis that the peptidase activity in the full-length enzyme outpaces that of ATPase activity. To explore the possibility that the inhibition of peptidase activity might be due to $Mg^{2+}$, we conducted a peptidase activity assay on the wild-type enzyme, both with and without the presence of $Mg^{2+}$ (Supplementary Fig. 1f). We observed that $Mg^{2+}$ had no discernible impact on the peptidase activity. Thus collectively, our findings suggest that the ATP hydrolysis cycle of ComA impedes its peptidase activity, implying that the peptidase and ATPase activities occur in a sequential manner, rather than concurrently.

Our observation that the activity of the PEP domain is likely not coupled with ATP hydrolysis because the residual peptidase activity of ComA with ATP was similar to that of the E647Q mutant (Fig. 1f and

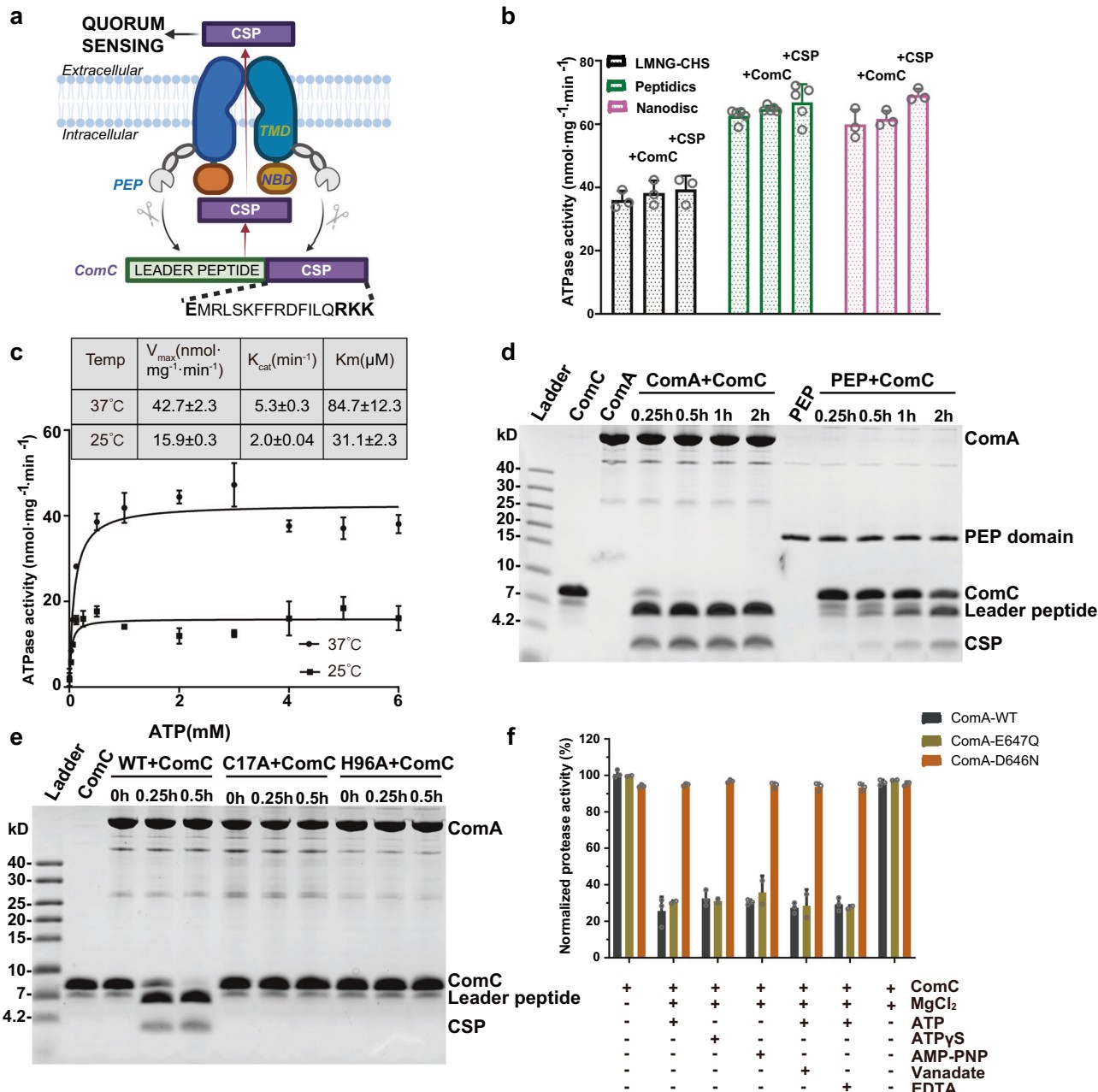

**Fig. 1 | Functional characterization of ComA. a** Diagram of ComA architecture and its role in CSP process and secretion for Quorum sensing. TMD: transmembrane domain; NBD: nucleotide-binding domain; PEP: peptidase domain. **b** ATPase activity of ComA in detergents (LMNG), peptidisc and nanodisc (MSP1D3) environment, in the presence or absence of substrates or product. Individual data points are presented as circles whereas standard deviation (SD) are shown as error bars ($n = 3$ for ComA in LMNG-CHS, $n = 5$ for ComA with peptidisc, $n = 3$ for ComA in nanodisc.). **c** The ATPase activity of ComA in detergents checked at different temperatures. SD shown as error bars. The experiments were repeated three times with biological replicates. **d** The peptidase activity of full-length ComA and truncated peptidase domain (PEP) checked at 25℃. The experiment was repeated three times independently with similar results. **e** The peptidase activity of wild-type ComA (WT) and ComA mutants measured at 25℃. The experiment was repeated three times independently with similar results. **f** ATP analogues inhibited the peptidase activity of ComA (WT) and E647Q mutant, but have no effect on D646N which lose the ability to bind ATP. Mg²⁺ alone have no effect on all ComA measured. Individual data points are presented as circles whereas SD are shown as error bars ($n = 3$ for all samples.).

Supplementary Fig. 1d). Similar findings were previously reported in PCAT1 study[28], where it was noted that the PEP domain is associated with the primary structure of the ABC transporter when the two NBD domains are isolated from one another. Concurrently, the ATP binding initiates the dissociation of the PEP domains, subsequently leading to their flexibility. We aimed to examine if this hypothesis is applicable to ComA; hence, a cryo-electron microscopy (cryo-EM) study of ComA mutants was conducted in the presence or absence of substrate CSP or ATP (Supplementary Fig. 2 and 3).

For the experiment, we utilized a ComA C17A mutant in the presence of ComC, as this mutant, devoid of protease activity, is predicted to trap the enzyme in a ComC-bound state. Similarly, an E647Q mutant was employed in the presence of ATP, considering that its lack of ATPase activity likely traps the enzyme in an ATP-bound state. Following careful cryo-EM analysis, we found that each sample disclosed a single conformation. The cryo-EM map of the sample with the peptide presented a resolution of approximately 6 Å, with only the secondary structure being delineated (Fig. 2a). In contrast, the

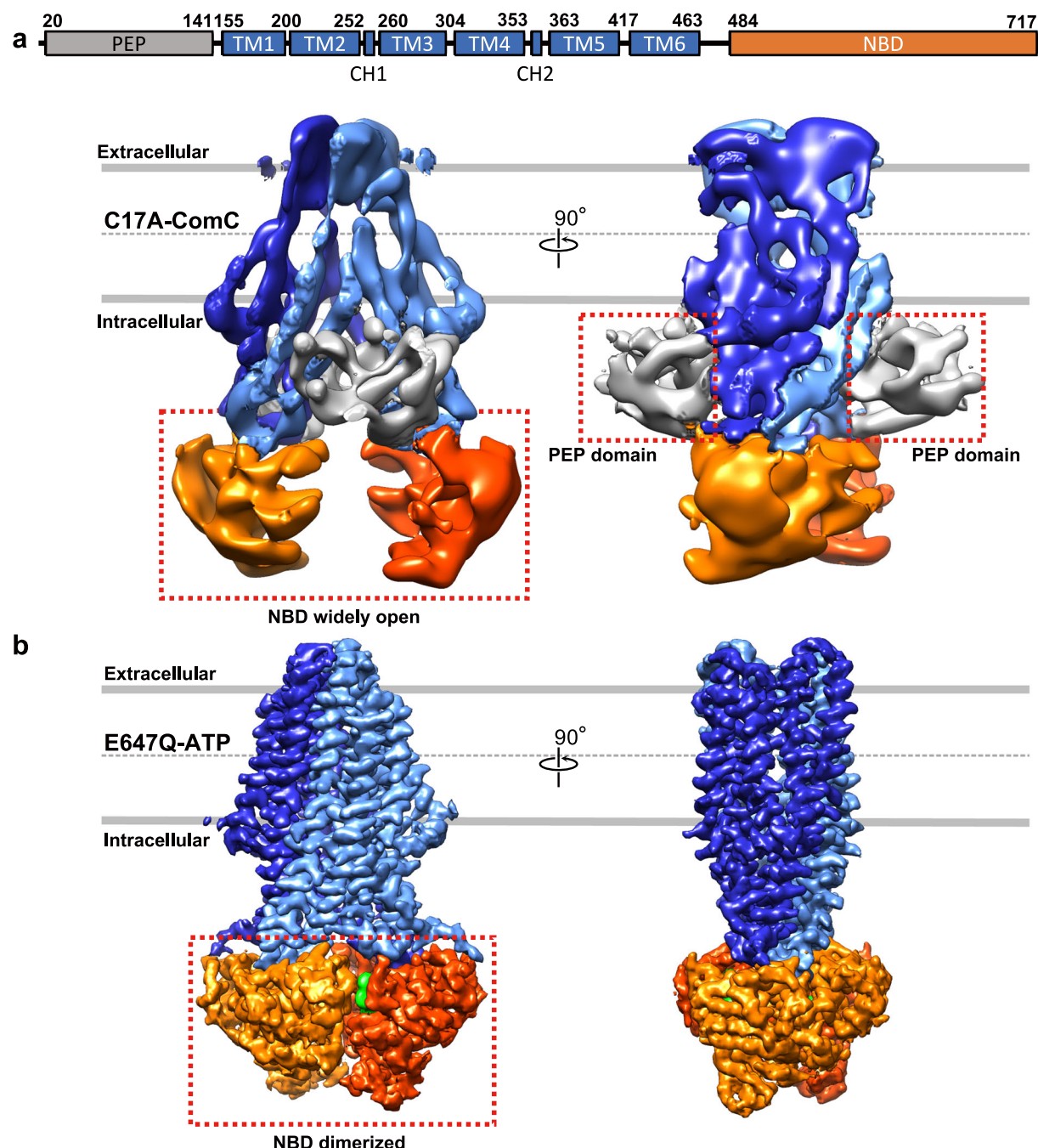

**Fig. 2 | Comparative structures of full-length ComA in ATP-bound and unbound states. a** (top), displays a cartoon depiction of the domain architecture of ComA, consisting of a peptidase domain (PEP domain), a transmembrane domain (TMD, encompassing TM1-TM6), and a nucleotide-binding domain (NBD) arranged from the N-terminus to the C-terminus. CH: coupling helices. **a** (bottom), presents both front and side views of the cryo-EM density map and model of the ComA C17A mutant with ComC, but lacking ATP. The cryo-EM map contour level is set to 0.004 in Chimera. **b**, shows the front and side views of the cryo-EM density map of the ComA E647Q mutant in the ATP-bound state, with the contour level of the EM map set to 0.019 in Chimera. In the ATP-absent state (panel **a**, bottom), the sample with ComC exhibits NBD domains that are widely separated, while the two PEP domains are ordered and associated with the core structure of the ABC transporter. Conversely, in the ATP-bound state, represented by the E647Q-ATP map (panel **b**), the two NBD domains dimerize upon ATP binding, whereas the PEP domain dissociates from the ABC transporter and exhibits increased flexibility. The color scheme used is as follows: the PEP domain is depicted in grey, the TMD in blue, the NBD in orange, and the two bound ATP molecules are shown as green spheres.

sample in the ATP environment achieved a higher resolution at 3.1 Å (Fig. 2b).

Interestingly, substantial conformational differences were detected within the PEP domain and NBD domains. In the peptide presence, a clear density of the two PEP domains was observed, positioned at the two intracellular gates that connect with the central cavity of the TMD

domain (Supplementary Fig. 2f and Fig. 2a). Simultaneously, the two NBD domains were found to be widely separated from each other. However, under ATP presence, the two NBD domains exhibited a close association, and the EM density of the two PEP domains disappeared (Fig. 2b), implying their dissociation from the primary ABC transporter structure and subsequent transition to a highly flexible state.

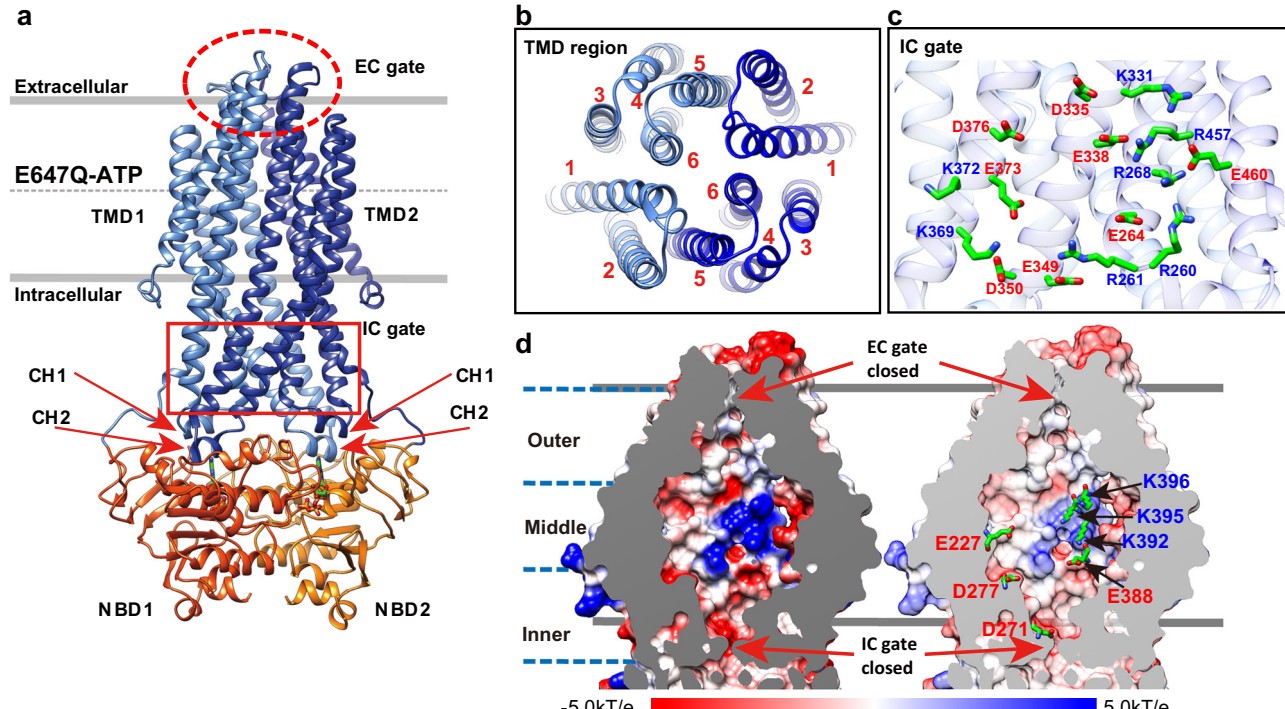

**Fig. 3 | The ATP-bound structure of ComA E647Q reveals distinctive electrostatic characteristics of the intracellular (IC) gate and central cavity. a** The overall structure of the ATP-bound ComA E647Q mutant is illustrated. The coupling helices, CH1 and CH2, play a crucial role in transferring ATP binding, hydrolysis, and release signals from the NBD domain to the TMD domain. **b** The dimeric interface of the ComA dimer at the TMD domain is displayed. Each ComA monomer's TMD domain comprises 6 transmembrane (TM) helices. **c** A detailed view of the IC gate region reveals a comprehensive electrostatic network. **d** An electrostatic potential calculation of the TMD cavity is shown. The surface potential representation ranges from −5kT/e (depicted in red) to +5kT/e (depicted in blue). The charge distribution was generated using the APBS program and visualized in Chimera. The color scheme is as follows: TMD is presented in blue, NBD in orange, and the two ATP molecules bound are shown in green.

To further confirm the dissociation of the PEP domain upon ATP binding, we conducted cross-linking mass spectrometry (CLMS) experiments on ComA in the presence and absence of ATP, cross-linked with disuccinimidyl sulfoxide (DSSO) with 2 biological replicates digested with trypsin and 2 biological replicates digested with chymotrypsin. Our CLMS analysis with 1% false discovery rate (FDR) cut-off identified a total of 46 unique cross-link sites across both states, with 27 cross-links found in ATP-free state only, 6 cross-links found in ATP-bound state only and 13 cross-links found in both ATP-free and ATP-bound states (Supplementary Fig. 4, Supplementary Table 1). Cross-links, which fall on the PEP region were mapped to the cryo-EM structure, with approximately 60% of cross-links within the expected Cα-Cα distance of 30Å for DSSO cross-links. Our CLMS results confirm the relative locations between the intracellular (IC) gate region and the PEP domain, including cross-links between residues 43/377, 43/380, 121/372, and 121/694 in the absence of ATP (Supplementary Fig. 4, Supplementary Table 1). While in the presence of ATP, only 1 cross-link between residues 121/372 was observed between the PEP domain and ICgate in the post-cleavage state (with ATP), compared to the 4 unique cross-links observed in the pre-cleavage state (without ATP) (Fig. S2, Table S1). Similar amounts of protein were analyzed for both ATP-free and ATP-bound states, with unique peptide counts of noncross-linked peptides of 48, 45 for ATP-free trypsin replicates; 45, 48 for ATP-bound trypsin replicates; 13, 15 for ATP-free chymotrypsin replicates and 11, 13 for ATP-bound chymotrypsin replicates. The lower number of cross-links observed between the PEP domain and ICgate in the ATP-bound state compared to the ATP-free state are likely to be due to the presence of ATP rather than the overall protein amount. Together, these findings suggest that the peptide processing step might be conserved among PCAT family members.

## Structure of peptide efflux pump unveiling unique electrostatic features of the IC gate and the central cavity

Our 3.1 Å structure of ATP-bound ComA (E647Q) mutant sample was in the absence of Mg²⁺. Two ATP molecules were found in the NBDs (Supplementary Fig. 5), the TMD and the two NBDs were well resolved and therefore allowed an unambiguous model building (Supplementary Fig. 5).

The overall architecture of ComA is a homodimer, similar to the other bacterial exporters like McjD[38,39], Sav1866[40], PCAT1[29], and MsbA[41–43] (Fig. 3a). The two monomers form a swapped dimer (Fig. 3a), with an interface composed of TM helices 1, 2, 3, and 6 from one monomer and TM helices 4 and 5 from the other monomer (Fig. 3b). The coupling helices (CH1/2), located between TM helices 2 and 3 (CH1) and 4 and 5 (CH2), interact with the NBDs and transmit conformational changes from the NBDs to the TMDs (Fig. 3a). CH1 interacts with both NBDs, while CH2 interacts with only one NBD from the opposite monomer. The overall structure of NBD domain resolved in the context of full-length protein is less compact, about 4 Å wider than that of previously determined NBD domain alone (PDB ID: 3VX4)[33] (Supplementary Fig. 6a). Dimerization of the NBD domains in the presence of ATP closes the IC gate (Supplementary Fig. 6b). Unlike most ABC transporters, where ATP binding induces the outward-facing (OF) conformation with an open extracellular (EC) gate, the EC gate remains closed in ATP-bound ComA. This OF-occluded conformation has been observed in PCAT1[29] and the antibacterial peptide transporter McjD[39].

Unlike other PCATs, ComA has a remarkable electrostatic network consisting of 30 charged residues distributed on both sides of its IC gate. This network, including R260, R261, E264, R268 in TM3; K331, D335, E338, E349, D350 from TM4; K369, K372, E373, D376 from TM5;

R457 and Glu460 from TM6 (Fig. 3c), creates a complex system of interactions that restricts the opening of the IC gate. This electrostatic network also maintains the two NBDs in close proximity, contributing to the unusually high ATP affinity to ATP compared to other ABC transporters. To verify this assumption, we measured the affinity of ATP to several ComA mutants with their electrostatic network disrupted. The single mutations (i.e. R260A, R261A, E264A) only led to a minor decrease in ATP affinity (a higher Km) compared to the WT enzyme (Supplementary Fig. 6c), while introducing more mutations (i.e. R457A/E460A) resulted in significant ATP affinity reduction. However, our attempts to mutate more residues (i.e. R268A/R457A/E460A, R261A/R457A/E460A) led to protein instability, contrary to the WT enzyme, which demonstrated high thermostability and it retained high activity under high temperatures for an extended period. As a result, our mutagenesis study confirmed the critical role of the electrostatic network's interactions in both ATP affinity and protein stability.

Our structural analysis, utilizing the 3V server[44], has also revealed a large occluded central cavity similar to the ATP-bound PCAT1 structure[29], with an estimated volume of 6612 Å$^3$, sufficient to accommodate at least two CSP molecules (Supplementary Fig. 7). As our study below suggests simultaneous translocation of two CSP molecules, ComA's cavity size situates between two peptide exporters with nucleotide-bound structures available: larger than McjD's (transporting a 21-amino acid substrate and having a cavity less than 5000 Å$^3$)[38], and smaller than PCAT1's (transporting a 66-amino acid substrate and having a cavity of 7958 Å$^3$)[28]. Though an apparent correlation between the central cavity size and substrate size may exist, it does not dictate a strict rule. For instance, smaller molecule transporters, such as rV1819c (transporting Bleomycin and vitamin B12)[45] and IrtAB (transporting the even smaller molecule, Carboxymycobactin)[46], have cavity sizes of 3780 Å$^3$ and 4600 Å$^3$, respectively. Thus, while cavity size might affect substrate accommodation, it's not always proportional to the size of the transported substrate.

In addition, this cavity also has a remarkable set of charged residues, which can be divided into three distinct parts: inner, middle and outer regions. The negatively charged inner region is closed to the IC-gate (Fig. 3d) and contains eight acidic residues (E227, D271, D277, and E388), all of which have the sidechain oriented towards the central channel. The positively charged lysine patch at the middle layer is composed of six lysine residues, three from each of the monomers (i.e. K392, K395, and K396). The outer part of the channel is predominantly hydrophobic. The peptide substrate CSP is highly electrostatic due to its positively charged C-terminus and a negatively charged N-terminus. Thus, we postulate that the inner and the middle layers may likely bind CSP through ionic interactions. However, these electrostatic interactions are presumably strong, which could pose a significant challenge to the subsequent release of CSP.

## A putative noncanonical binding site of CSP inside the translocation channel

We then performed cryo-EM experiments with ComA and ComC to understand the interaction between the mature CSP peptide and ComA (Supplementary Fig. 8). As ATP inhibits the peptidase activity, it was omitted from the reaction mixture. Under this condition, ComC would be hydrolyzed into CSP and likely loaded into the TM conduit. An EM density map at a resolution of 3.8 Å was obtained and the map was not affected by the symmetry applied in the final refinement (Supplementary Fig. 8g).

Indeed, we detected EM densities within the central cavity, which align with the presence of two CSP molecules (Fig. 4a). Although the resolution might not be sufficient for definitive assignment to CSP, cross-validation from multiple EM maps in this study derived from the same protein batch in differing functional states provided compelling evidence. These maps revealed no similar density at the equivalent location, effectively ruling out the presence of lipids or other molecules and supporting the identity as CSP. Drawing upon insights from the PCAT1 study[28,29], we postulate that the C-terminal of the peptide is first to enter the central cavity, leading to an orientation with the C-terminal facing towards the extracellular side. This supposition is further supported by our following mutagenesis study (discussed below) and the observation that 13 out of the 17 residues of the CSP were fitted acceptably into the density (Fig. 4a), which aligns with the hypothesis of ComC's leader peptide being cleaved. Collectively, these insights strengthen the plausibility of our proposed model. However, it is crucial to acknowledge that the EM density observed was not of sufficiently high resolution to definitively assign the bound CSP sequence. Therefore, while this model informs our understanding and is discussed in the current section, we have prudently opted for a poly-alanine model in the deposited structure to accommodate the inherent uncertainty.

The electron densities that are presumably corresponding to two CSP molecules are located at the hydrophobic outer region of the channel (Fig. 4a). Our modeling results suggest that the positively charged C-terminus of CSP is facing outward, away from the patch of lysine residues at the middle region. The two presumed CSP molecules are tightly packed against each other in an elongated orientation perpendicular to the membrane, and the interaction is dominated by hydrophobic contacts of non-polar residues along the substrate binding pocket (Fig. 4b). Specifically, several aromatic residues of CSP, including F7 and F11, are likely involved in the interaction. Three polar residues of ComA (Y216, Y433, and N436) are close to the presumed CSP electron densities (<4 Å) and may contribute to the binding. Furthermore, there is a methionine-aromatic interaction in the structural model between ComA/Y216 and CSP/M2.

As we could not clearly visualize the two CSPs inside the putative binding site, we asked whether mutations of these residues affect the peptidase activity of ComA. The rational is that if the putative CSP binding site within the channel is disrupted, the matured peptide would likely be entrapped within the central cavity. Given that the NBDs are already in close proximity, as observed in the CSP-bound structure, ATP binding and hydrolysis could occur normally. Simultaneously, the complete disappearance of EM density in CSP-bound structure and in our previous determined ATP-bound structure, indicates that the two PEP domains have dissociated from the ABC transporter. Therefore, the PEP domain is expected to behave similarly to the truncated version, exhibiting a reduction in peptidase activity. Indeed, Y216A and Y433A decreased the peptidase activity of ComA (Fig. 4c). Importantly, ComA(Y216A) and ComA(Y433A) have similar ATPase activities compared to the wild-type, indicating that these mutations did not affect the overall structure of ComA as expected (Supplementary Fig. 9). The putative CSP-binding pocket also contains two patches of residues with opposite charges: D194 and K392/K395/K396 (Fig. 4d). The positively charged lysine residues are located at the center region of the lumen near the N-terminal E1 of CSP, which is one of the two negatively charged residues of ComC. Notably, no electrostatic interaction is detectable between these residues. This may potentially facilitate the release of CSP. Instead, a potential salt bridge is formed between D194 of ComA and R9 of CSP based on our current model (Fig. 4b). This strong electrostatic interaction may help retain the two CSP molecules in the substrate binding pocket. Consistently, D194A markedly reduced the peptidase but not the ATPase activity of ComA (Fig. 4c and Supplementary Fig. 9). Although D199 is located near D194, it appeared to be unrelated to the CSP-ComA interaction because D199A had no detectable effect on the peptidase activity unless D194 was also mutated. Based on the biochemical evidence and the cryo-EM model, we suggest that CSP binds to the outer region of the conduit.

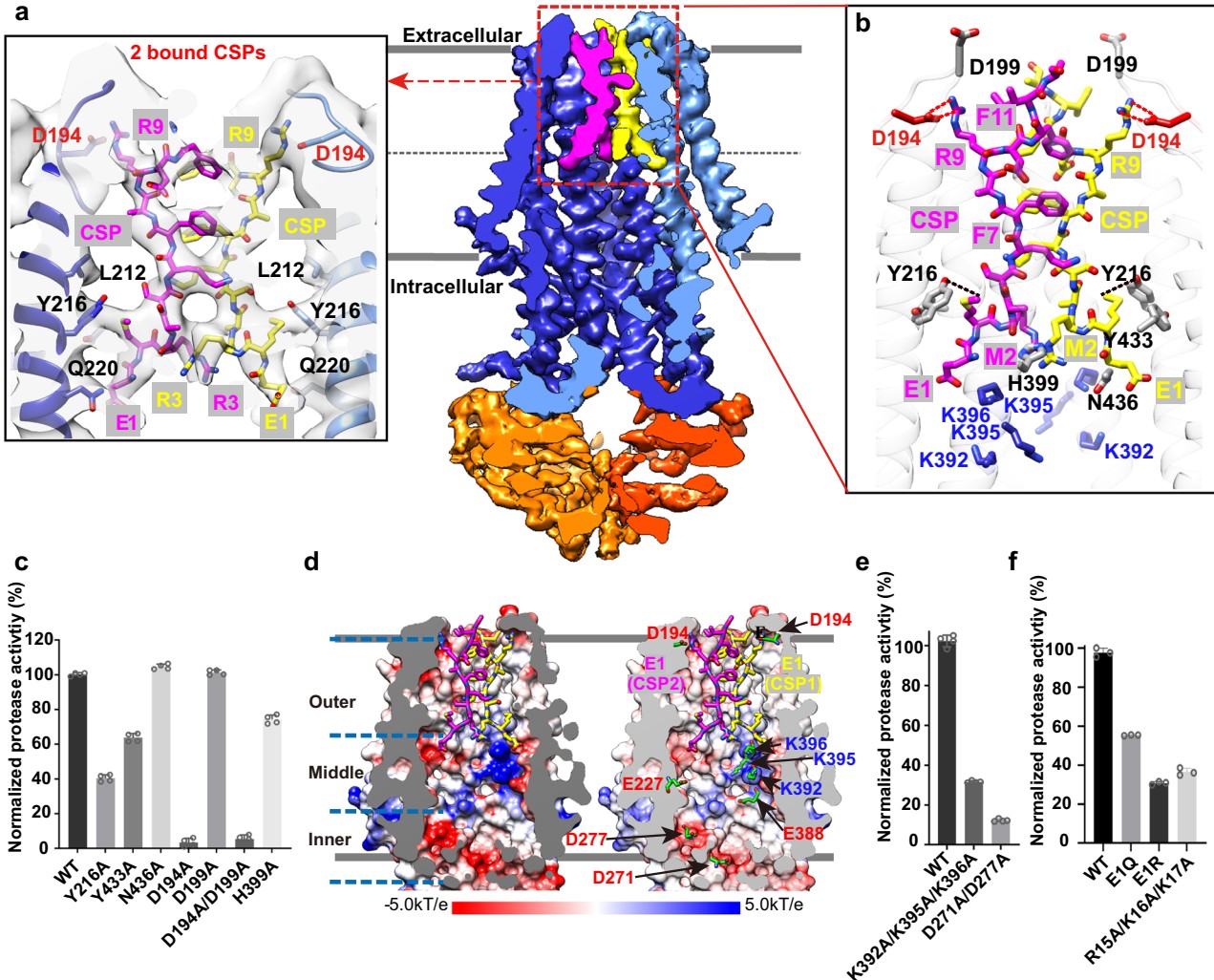

**Fig. 4 | Structural-Functional examination of putative CSP binding by ComA.**
**a** Cryo-EM density maps of ComA with two putative bound CSP molecules. On the left, a zoomed-in view shows the proposed CSP model fitted into the EM density map, with a contour level set to 0.014 in Chimera. On the right, the EM density is color-coded with surface carving: magenta/yellow for CSP, blue for the TMD, and orange for the NBD. The coloring is carved based on the model with a surface zoning radius of 2.8 Å at a contour level of 0.014 in Chimera. **b** Illustrates potential interactions between ComA and CSP. Only residues within 4 Å of the ligand are shown. **c** Depicts the peptidase activity of mutant residues potentially involved in CSP binding. Individual data points are presented as circles whereas standard deviation (SD) are shown as error bars ($n = 4$ for all samples.). **d** Shows an

electrostatic potential calculation of the CSP binding pocket. The surface potential representation ranges from −5kT/e (red) to +5kT/e (blue), as generated using the APBS program and visualized in Chimera. **e** Displays the peptidase activity of ComA mutants with charged residues in the inner layer crucial for CSP translocation across the membrane. Individual data points are presented as circles whereas SD are shown as error bars ($n = 4$ for all samples.). **f** Demonstrates the peptidase activity of ComA in the presence of terminal charge residue mutants from ComC. The error bars in kinetic experiments represent the mean value +/- standard deviation (SD) for the group. All kinetic experiments were replicated three times with biological duplicates. Individual data points are presented as circles whereas SD are shown as error bars ($n = 3$ for all samples.).

## A model for translocating CSP across the cytoplasmic membrane

Cargo release in PCATs is thought to be triggered by proteolytic cleavage that frees the substrate from the PEP domain[28]. Yet, the driving force to move the cargo from the inner to the outer region of the channel remains unclear. We propose that the negatively charged inner region may repel the N-terminus of CSP (Fig. 4d). To test this hypothesis, we mutated D271, D277, K392, K395, and K396 of ComA. Disruption of the negative charge near the IC gate (D271A and D277A) reduced ComA peptidase activity by 5-fold. Moreover, mutations that reduce the positive charge in the middle region (K392A, K395A, and K396A) led to a decrease in the peptidase activity by approximately 60% (Fig. 4e). Again, these mutations did not disrupt the overall fold of ComA because they did not affect the ATPase activity (Supplementary Fig. 9). Thus, the charged

residues in the central lumen of ComA are likely important for CSP translocation.

Next, we tested whether mutations in E1, R15, K16, and K17 of CSP would cause a similar reduction in ComA peptidase activity. As expected, E1Q and E1R led to a defective peptidase activity in ComA, with E1R exhibiting a more severe reduction (Fig. 4f). The R15A/K16A/K17A triple mutant had a similar effect, confirming the importance of the C-terminal RKK motif. We could not fully exclude the possibility that the reduction in ComA peptidase activity was caused by an alteration in the ComC sequence. However, mutations of two charged regions produced similar results. Furthermore, we show that charged residues in both ComA and ComC are required for effectively processing the leader peptide. Together, these observations are consistent with a mechanism that CSP is pushed through the channel through charge repulsion.

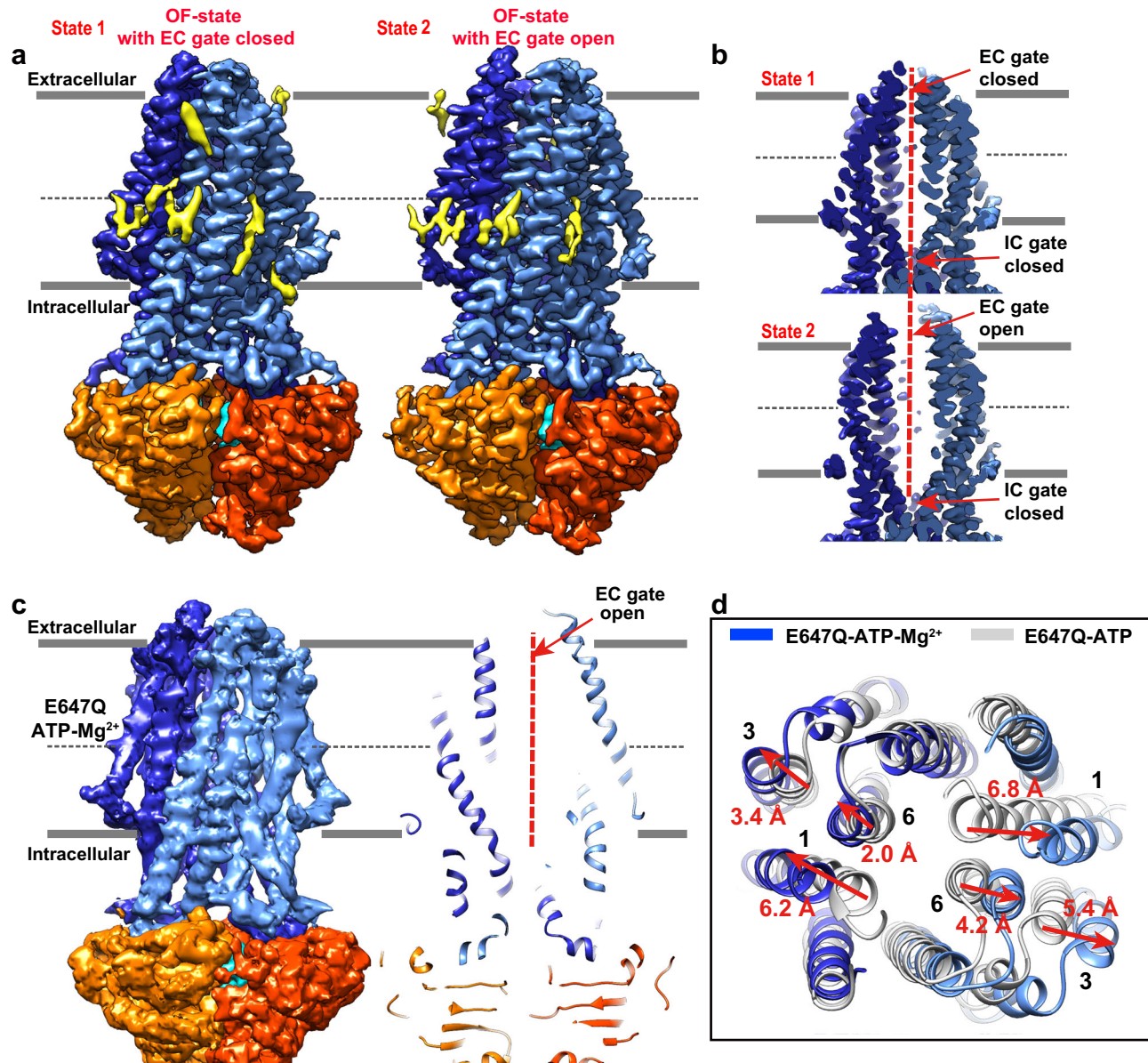

**Fig. 5 | Necessity of Mg²⁺ for opening of the EC gate / release of CSP from ComA.** **a** Surface views of cryo-EM density maps from two distinct states obtained by introducing ATPγS -Mg²⁺ to the CSP-bound sample. Bound lipids, shown in light yellow, vary between the two states. Both EM map contour levels are set to 0.027 in Chimera. **b** Cross-sectional views of EM density maps of the TMD from the two differing states. State 1 assumes an OF-state with a closed EC gate, whereas State 2 adopts an OF-state with an open EC gate. **c** Cryo-EM structure of ComA E647Q in the presence of ATP-Mg²⁺. On the left, the density map of ComA E647Q bound to ATP-

Mg²⁺ is displayed. On the right, a cross-sectional view of the ComA E647Q-ATP-Mg²⁺ model highlights the EC also in an open state. The contour level of the EM maps is set to 0.015 in Chimera. **d** Structural comparison between ComA E647Q in the presence of ATP only (colored in grey) and in the presence of both ATP and Mg²⁺ (colored in blue). Only the co-binding of ATP and Mg²⁺ initiates a transition of the EC gate from a closed to open state, while ATP alone does not instigate this change. Color scheme: TMD in blue, NBD in orange, two bound ATP molecules in cyan.

## Mg²⁺ is required for the release of CSP from ComA

ATP binding alone in our E647Q mutant structure traps ComA at an OF-occluded state, we thus speculated that a divalent cation like Mg²⁺ may be necessary for substrate release by opening the EC-gate. To test this, we conducted cryo-EM analyses on CSP-bound ComA in the presence of ATPγS-Mg²⁺ (Supplementary Fig. 10). Two conformations were observed with distinct extracellular TMD openings, both had a resolution of approximately 2.9 Å. The bound ATPγS and sidechains from both the NBD and TMD domains were clearly visible (Supplementary Fig. 11) but CSP density was absent in state 1 and weak in state 2,

indicating the bound CSP captured in our CSP-bound structure was released from ComA.

State 1 represented the OF-state with the EC gate closed, which is identical to the OF-occluded structure of ATP-bound ComA (E647Q) mutant (Figs. 5a and b). However, considering that the sample used here is derived from the same one that we obtained CSP-bound structure, while the closure of the EC gate, as well as the disappearance of two bound CSP molecules, is only observed in this study following the addition of ATPγS, we suggest that this state likely represents a post-translocation state that precedes ATP hydrolysis.

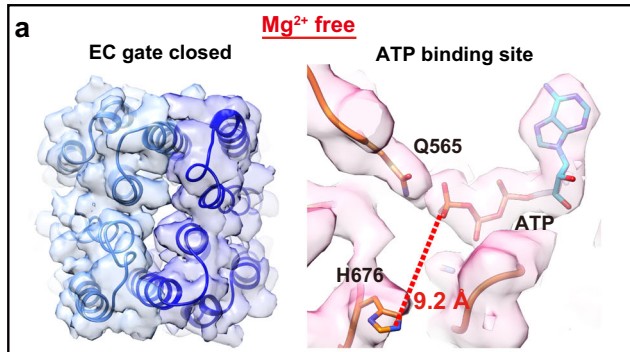
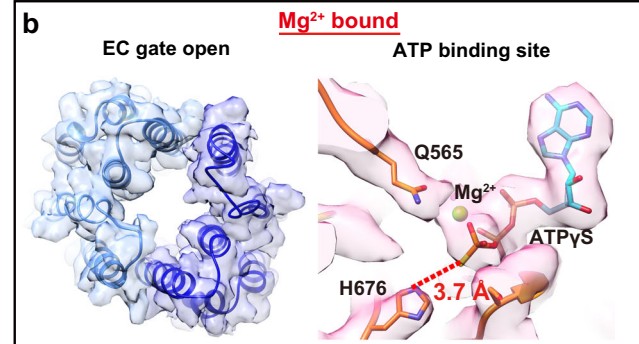
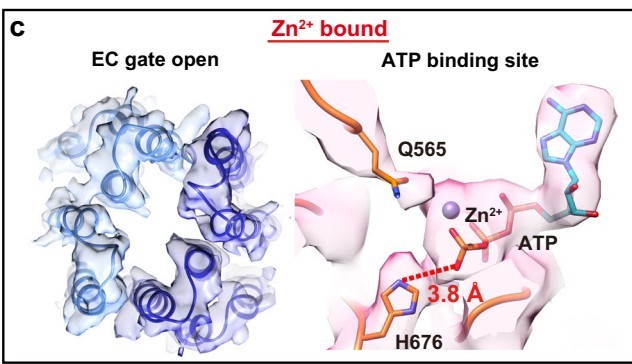
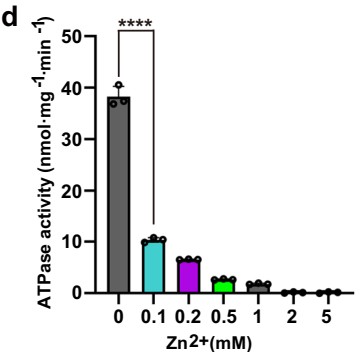

**Fig. 6 | His676 (H676) interaction with ATP's gamma phosphate, stabilized by a divalent cation, potentially drives EC gate opening for CSP release. a** Structure of the EC gate and ATP-binding site in the ATP-only bound E647Q mutant structure. Without $Mg^{2+}$, His676 is distant from the gamma phosphate (9.2 Å). The EM map contour level is set to 0.013 in Chimera. **b** Structure of the EC gate and ATP-binding site in the ATPγS-$Mg^{2+}$-bound structure. With $Mg^{2+}$, His676 forms a salt bridge with the gamma phosphate (3.7 Å). The EM map contour level is set to 0.017 in Chimera. **c** Structure of the EC gate and ATP-binding site in the ATP-$Zn^{2+}$-bound structure. In the presence of $Zn^{2+}$, which assumes the role of $Mg^{2+}$, His676 forms a salt bridge with the gamma phosphate (3.8 Å). The EM map contour level is set to 0.17 in Chimera. Color scheme: TMDs in blue, NBDs in orange, ATP in cyan, $Mg^{2+}$ in light green, and $Zn^{2+}$ in grey. **d** $Zn^{2+}$ inhibits ComA's ATPase activity. Individual data points are presented as circles whereas standard deviation (SD) are shown as error bars ($n = 3$ for all samples.). Statistical significance was assessed by an unpaired, two-tailed t-test, ****$P < 0.0001$.

State 2 drew our attention because it was in an OF-conformation with the EC gate open, which may allow the bound CSPs to escape (Figs. 5a and b). A dim density becomes discernible at the N-terminal region of bound CSP, coinciding with the CSP-binding site. This density, however, appears considerably weaker than that observed in the CSP-bound structure. We propose that this may be attributed to a fraction of particles in the sample demonstrating a partial release of CSP. The IC gate remained closed to prevent the substrate from diffusing back to the cytosol (Fig. 5b). The major difference between these two states was in the outer portion of the channel, where large shifts occurred in TM helices 1 and 6, resulting in an open EC gate (Supplementary Fig. 12). Additionally, the pattern of lipids bound to the TM helices was different (Fig. 5a). We further compared the OF EC-gate open state with the CSP-bound structure (Supplementary Fig. 13). Overall, the conformational changes are minor (RMSD 1.9 Å) (Supplementary Fig. 13a) and are concentrated at the putative CSP binding pocket (Supplementary Fig. 13b). First, the sidechains of the two D194 residues rotate away from the CSP upon ATPγS-$Mg^{2+}$ binding, which was further stabilized by hydrogen bonds with residue S420 in TM6. As D194 is likely crucial for CSP binding, its movement may disrupt the interaction between ComA and CSP. Moreover, TM2 squeezes inward by about 2 Å toward the central translocation pathway, which may destabilize the CSP binding pocket.

To verify that the binding of ATP-$Mg^{2+}$, rather than ATP alone, instigates the opening of the extracellular (EC) gate, we determined a structure of the ComA (E647Q) mutant in the presence of both ATP and $Mg^{2+}$ (Supplementary Fig. 14). This was based on the previously resolved E647Q mutant sample in the presence of ATP alone, which demonstrated a closed EC gate (Fig. 3). The structure, at approximately 4.5 Å resolution, reveals the EC gate in an unequivocally open conformation (Fig. 5c), markedly divergent from the structure observed in the presence of ATP alone (Fig. 5d). Although the resolution of this structure is relatively low, the stark contrast in the state of the EC gate between the two conditions provides compelling evidence of the crucial role that $Mg^{2+}$ plays in facilitating the opening of the EC gate and the subsequent release of CSP. This comparative study, using the same mutant protein, thus strengthens our understanding of the interplay between ATP-$Mg^{2+}$ binding and EC gate functionality.

**ATP-$Zn^{2+}$ trap ComA in the OF, EC gate open state**

To explore the role of $Mg^{2+}$ in triggering the opening of the EC gate for CSP release, we conducted a comparative analysis of the ATP binding site in two structures, both resolved at approximately 3 Å: the ATP-only bound E647Q mutant (Fig. 6a) and the ATPγS-$Mg^{2+}$-bound structure (Fig. 6b).

While the overall architecture of the ATP-binding site remains largely conserved between the two structures, we noticed that the residue H676, positioned adjacent to the gamma phosphate of ATP, rotates inward in the ATPγS-$Mg^{2+}$-bound structure. This residue likely interacts with the gamma phosphate, thereby stabilizing the bound $Mg^{2+}$. Such a similar stretched conformation of H676 was also observed in the previous structure of NBD domain alone resolved in the presence of ATP-$Mg^{2+}$ (PDB ID: 3VX4) (Supplementary Fig. 15) This observation suggests a plausible mechanism by which ATP binding alone is inadequate for facilitating EC-gate opening and CSP release. Instead, the presence of $Mg^{2+}$ appears to be necessary to induce the requisite conformational changes for this process.

To test if other divalent cations could trigger the EC-gate to open and confirm the proposed mechanism from above, we solved the structure of ComA by cryo-EM in the presence of $Zn^{2+}$, which resulted in a single conformation with a resolution of ~3.9 Å. This conformation represents the OF-state with the EC gate open (Supplementary Fig. 16 and 17a). The $Zn^{2+}$ ion was found between the gamma-phosphate of ATP and the Q565 residue (Fig. 6c), exactly replacing the $Mg^{2+}$. The overall conformation was similar to the ATPγS-$Mg^{2+}$-bound state, with slight differences observed at the TM helices at the EC gate (Supplementary Fig. 17b). Additionally, in the $Zn^{2+}$-bound state, the H676 residue also take a stretched conformation that rotate toward the gamma phosphate group of ATP and formed a hydrogen bond, similar to that observed in the ATPγS-$Mg^{2+}$-bound structure (Fig. 6c).

$Zn^{2+}$, known to contain high toxicity to pathogens[47], serves as a defensive strategy employed by plants[48], though its precise mechanism remains ambiguous. The geometric parameters of $Zn^{2+}$ interactions differ from those of $Mg^{2+}$ [49,50], which holds significant implications for ATP hydrolysis, a process known to involve the nucleophilic attack of an activated water molecule at the ATP γ-phosphate[51]. The binding of this water molecule is influenced by the bound metal. Therefore, $Zn^{2+}$, with its distinctive structural chemistry compared to $Mg^{2+}$, is likely to disrupt this step, functioning effectively as an inhibitor. Indeed, $Zn^{2+}$ strongly inhibited the ATPase activity of ComA (Fig. 6d). Consistent with the postulation that the $Zn^{2+}$-bound state is non-functional but stable, almost all particles in the cryo-EM experiment were trapped at the OF state with the EC gate open.

Collectively, these findings suggest that other divalent metals, such as $Zn^{2+}$, could supplant $Mg^{2+}$ at the ATP binding site and maintain a comparable overall conformation. As no significant conformational changes occur, the inhibitory effect of $Zn^{2+}$ likely originates from its unique interaction geometry with water molecules, which differs from that of $Mg^{2+}$. Intriguingly, both $Zn^{2+}$ and $Mg^{2+}$ binding triggers the opening of the EC gate. This observation not only validates the role of $Mg^{2+}$ binding in the EC gate opening, but also corroborates the hypothesized mechanism, likely mediated through the conformational change of His676 at the ATP binding site.

### Validation of residues involved in CSP secretion

We next validated the critical residues of ComA and ComC by measuring CSP export in *S. pneumoniae* using the HiBiT assay[52]. First, *comC* was fused with a HiBiT tag at the native locus (*comC*-HiBiT). The bacteriocin transporter BlpA was then inactivated to prevent its interference in the assay[52]. Subsequently, the corresponding *comA* mutants were constructed in the Δ*blpA comC*-HiBiT background. *comC-HiBiT* expression was induced by adding CSP to the culture. If ComC-HiBiT was processed and exported by ComA or its variants, the CSP-HiBiT fusion would be detected in the culture supernatant when LgBiT and a luminescent substrate were added. Otherwise, the ComC-HiBiT peptide would be trapped in the cytoplasm. Thus, the CSP-HiBiT fusion could only be detected in the cell pellet fraction.

As expected, changing residues Y216 and Y433 in ComA to alanine reduced CSP export by ~2-fold, likely because these variants are defective in CSP-binding (Figs. 4c, 7a, and b). Likewise, residue D194 was shown to be important for CSP export (Figs. 7a and b) as it is part of the negative center near the EC gate, which is presumably crucial for CSP binding and release. By contrast, consistent with the biochemical assays (Fig. 4c), D199A had a relatively small effect on CSP export (Figs. 7a and b). Similarly, disruption of the charged residues in the lumen (D271A/D277A and K392A/K395A/K396A) resulted in a significant reduction in CSP secretion, supporting the hypothesis that the opposite charges may be a driving force for CSP translocation (Figs. 7a and b). We also tested two other residues (H399 and N436) that may interact with CSP but they are dispensable for ComA

activity in vivo. Next, to further illustrate the importance of electrostatic interactions between ComA and ComC in CSP export, we mutagenized the charged residues of ComC at its termini (i.e. E1Q, E1R, R15A/K16A/K17A, and ΔR15-K17). E1Q and E1R expectedly abolished ComC transport (Figs. 7c and d). Intriguingly, CSP export seemed to be unaffected when the positive charge of the CSP C-terminus was removed (Figs. 7c and d). These residues are nevertheless important for the expression or stability of CSP, as the ΔR15-K17 mutant had a significant decrease in CSP levels in the cell (Figs. 7c and d). Last, co-expressing the soluble PEP domain with ComC and the ComA variant that is defective in peptidase did not rescue CSP export (Supplementary Fig. 18). This result indicates that the peptidase activity is coupled to the transport function. In conclusion, the critical residues identified in our structural study were validated by directly measuring CSP export in *S. pneumoniae*.

## Discussion

In this study, we report the structures of the CSP efflux pump ComA in the presence or absence of ATP, ATPγS-$Mg^{2+}$, ATP-$Mg^{2+}$, ATP-$Zn^{2+}$, and ComC. ComA has several distinct features, including an extensive electrostatic interaction network at the IC gate. This may account for the unusually high ATP affinity and thermostability, perhaps important for enabling robust QS in an ever-changing environment. Moreover, the inner and middle regions of the central cavity are charged, which may push the CSP molecules through the channel after leader peptide cleavage. The remodeling of the central translocation tunnel may also help CSP export (Fig. 7). We also identified a putative CSP binding site at the outer region of the conduit. This binding site can presumably accommodate two CSP molecules even in the absence of ATP. For ComA, the OF conformation is induced by CSP or nucleotide binding. However, it seems to be insufficient to open the EC gate or trigger the release of CSP. To this end, ATP-$Mg^{2+}$ rather than ATP stably traps ComA at an OF-state with an open EC gate. Thus, divalent cations, such as $Mg^{2+}$ in the ATP-$Mg^{2+}$ complex, are required for conformational changes to open the EC gate. Together, these unique features of ComA underscore the importance of studying a diverse set of ABC transporters within the same family. Although they may overall adopt the universal alternative-access mechanism for substrate transport, there could be substantial structural differences that imply a variation in their transport mechanism.

Based on the ComA structures trapped in different states and on the assumption that the peptide processing mechanism is likely conserved among PCATs, which has been elegantly studied in PCAT1[28,29], we propose a working model (Fig. 8) for the processing and secretion of CSP in quorum sensing, which consists of the following stages: 1) Initially, as seen in apo PCAT1[29], the two PEP domains are located around the intracellular gate, while the two NBD domains remain separated from each other. 2) Then, in a process analogous to our presumed ComC-bound structure, which strongly mirrors the peptide-bound PCAT1 structure[28], the substrate peptide binds to the PEP domain located at the intracellular site. 3) After cleavage of the leader peptide, the two mature CSP molecules enter the central cavity, a state captured in our ATP-absent CSP-bound ComA structure. The negatively charged residues in the cytosolic region of the conduit facilitate the translocation of CSP molecules towards the outer leaflet binding site. Two potential salt bridges between D194 (ComA) and R9 (CSP) stablize the substrate binding. Once CSP binds, the intracellular gate closes, the PEP domains dissociate from the ABC transporter, and the two NBDs approach each other, preparing for ATP binding. 4) Subsequently, ATP and $Mg^{2+}$ bind to the two NBDs, causing local reconfiguration of the putative CSP binding pocket, including disruption of the potential D194 (ComA) - R9 (CSP) ion pair, and minor shifts in TM2. The opening at the extracellular gate, with a radius larger than 5.2 Å, enables the release of CSP from the central cavity. 5) Finally, after the CSP molecules are released, $Mg^{2+}$ dissociate from the ATP binding site,

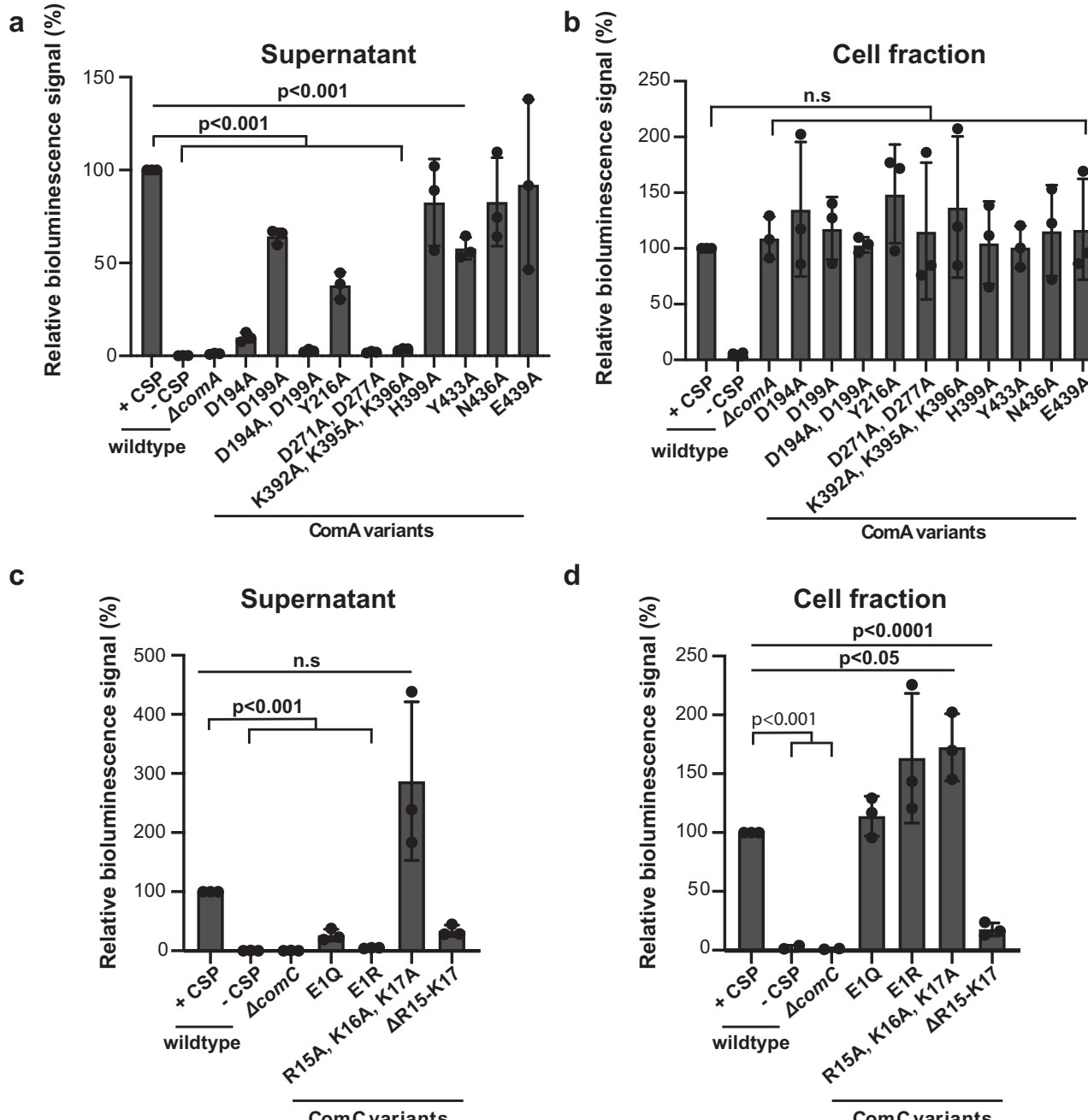

**Fig. 7 | In vivo validation of Key residues identified for ComA function a-b, Key residues identified for ComA function from structural and in-vitro biochemical studies.** In the supernatant, our statistical analysis reveals the p-values are all below 0.001: 1) Between wildtype ComA with CSP (+CSP) and without CSP (-CSP); 2) Between wildtype ComA in the presence of CSP (+CSP) and the ComA triple mutant K392A, K395A, K296A; and 3) Between wildtype ComA in the presence of CSP (+CSP) and the ComA mutant Y433A. In cell fraction, our statistical analysis reveals the following p-values: 1) Between wildtype ComA with CSP (+CSP) and comA knowckout (ΔcomA) is 0.4328; 2) Between wildtype ComA in the presence of CSP (+CSP) and the ComA E439A is 0.5469. When p-values exceed 0.05, it is indicated as "n.s.," the observed differences are statistically non-significant. **c–d** Role of charged residues from ComC in CSP export. Individual data points are presented as circles whereas standard deviation(SD) are shown as error bars ($n = 3$ for all samples.). In the supernatant: The $p$ value between ComA with CSP (+CSP) and ComA without CSP (-CSP) is <0.001; The p-value between ComA with CSP (+CSP) and the ComC mutant E1R is <0.001; the $p$ value between ComA with CSP (+CSP) and ComC mutant R15A, K16A, K17A is 0.0734. In the cell fraction: The $p$ value between ComA with CSP (+CSP) and ComA without CSP (-CSP) is <0.001; The $p$ value between ComA with CSP (+CSP) and ComA with comC knockout (ΔcomC) is <0.001; The $p$ value between ComA with CSP (+CSP) and ComA with the comC triple mutant R15A, K16A, K17A is <0.05; The $p$ value between ComA with CSP (+CSP) and ComA with the comC mutant ΔR15-K17 is <0.0001. Again, when p-values exceed 0.05, they are denoted as "n.s." to indicate that the observed differences are statistically non-significant. Statistical significance was analyzed by two two-tailed unpaired t test.

the extracellular gate closes once more, resulted in an OF-occluded state as observed in PCAT1 as well[29]. This conformation with Mg²⁺ dissociated may be facilitated by ATP hydrolysis in cell, followed by ADP release from the NBD domain. ComA then reverts to the apo state, ready for the next cycle of CSP processing and secretion.

One limitation of this study is that the PEP domains of ComA could not be clearly resolved to high resolution. The reason why these domains are especially flexible in ComA is unclear, but it may be due to the transient fashion of CSP processing. Future directions will be to trap the pre-cleavage state using noncleavable ComC analog and to

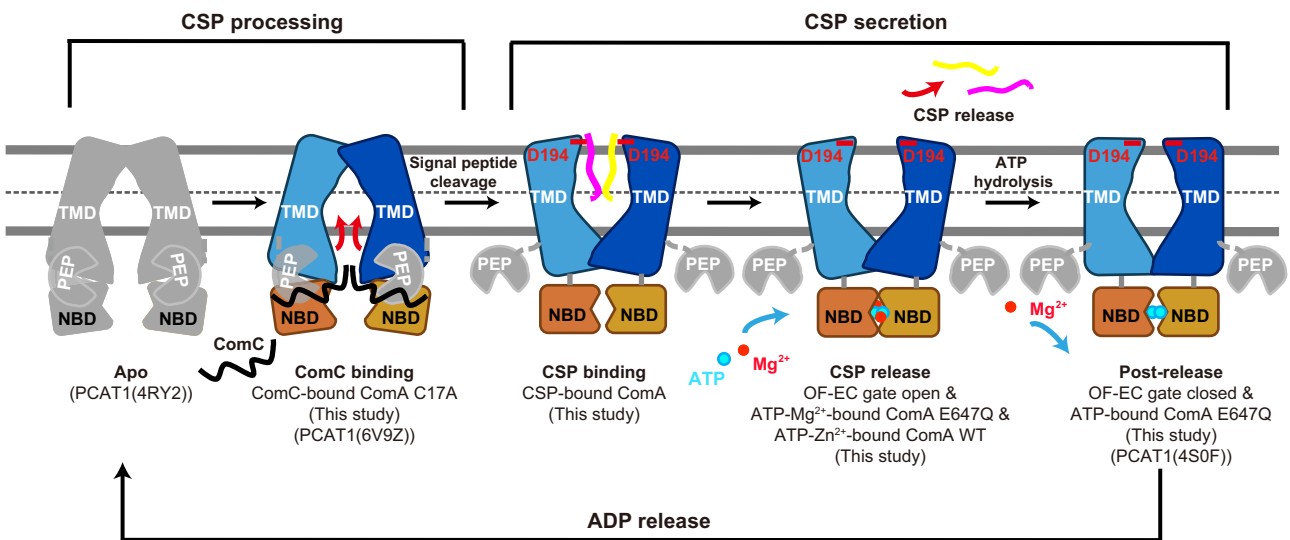

**Fig. 8 | Working model of ComA functionality in CSP processing and secretion.** The model's progression is detailed through different functional states, with snapshots from the relevant structures, as indicated beneath each state. Information for the CSP processing section (the apo state and ComC binding state) is primarily derived from previous PCAT1 study. 1) In the initial **apo** state, as seen in the apo PCAT1 structure (PDB ID: 4RY2), the two PEP domains are situated around the IC gate, while the two NBD domains remain separate. 2) Then for **ComC binding**, similar to our presumed ComC-bound structure, which closely resembles the peptide-bound PCAT1 structure (PDB ID: 6V9Z), the substrate peptide attaches to the PEP domain at the IC gate region, allowing the matured CSP to enter the central cavity with C-terminal enter first. 3) Following leader peptide cleavage, the two mature CSP molecules enter the central cavity for **CSP binding**, as depicted in our ATP-absent, CSP-bound ComA structure. Substrate binding is stabilized by potential salt bridges between D194 (ComA) and R9 (CSP). Post CSP-binding, the intracellular gate closes, the PEP domains disengage from the ABC transporter, and the two NBDs come closer, ready for ATP binding. 4) Then, ATP and $Mg^{2+}$ bind to the two NBDs, causing the extracellular gate to open for **CSP release**. 5) Finally, after CSP release (**post-release** state), the extracellular gate closes, ComA adopts an occluded conformation as previously captured in PCAT1 (PDB ID: 4S0F). Followed by ADP release from the NBD domain, ComA reverts to the apo state, primed for the next cycle of CSP processing and secretion. Color scheme: TMDs in blue, NBDs in orange, the two CSPs are individually color-coded in magenta and yellow, ATP/ADP is cyan, $Mg^{2+}$ is red, and the corresponding structures (from both this study and previous PCATs study) captured for each functional state are detailed. Unresolved structures in this study are depicted in grey.

investigate how ATP binding inhibits the peptidase activity of ComA. Nevertheless, the peptidase activity seems to couple with the transporter function (Supplementary Fig. 18). Additionally, the binding of CSP at the noncanonical CSP binding site likely feedback to the PEP domains, such that lesions in this region reduce the peptidase activity of ComA. We believe that this binding site holds great potential as a drug target since it is surface exposed. Efforts to engineer CSP are underway (REF) that harness the unique electrostatic feature of the CSP for custom inhibitor design. Inhibitors targeting CSP binding to ComA may be developed as anti-virulence molecules to combat gram-positive pathogens.

Several questions regarding ComA's mechanism remain unanswered. Firstly, it is unclear whether two tightly packed CSP molecules are secreted together at the same time under physiological conditions. Secondly, whether ComA has a "wide-opening" OF conformation similar to other ABC transporters remains elusive. The bound CSPs apparently are in a stretched orientation, so the "semi-opened" state of the EC gate may already be sufficient for export. However, we cannot exclude the possibility that a full-opening state transiently exists. Finally, the role of ComB in CSP transport remains unclear. Although ComB was thought to be part of the CSP transporter, we have yet to detect any interaction between ComB and ComA in vitro. Future studies will focus on revealing these missing links and developing ComA inhibitors for therapeutics.

## Methods
### Cloning, expression, and purification of ComA (mutants) and ComC
The ComA, truncated peptidase domain (PEP) of ComA, and ComC genes were amplified from the genomic DNA of *Streptococcus pneumoniae* (D39 strain), and cloned into the pRSFDuet-M2 or pET-15b vector with an MBP tag or His tag at N-terminus. Vectors of ComA mutants (Y216A, Y433A, N436A, H399A, D194A, D199A, D194A/D199A, D271A/D277A and K392A/K395A/K396A) and ComC mutants (E1Q and E1R) were constructed using site-directed mutagenesis method.

For ComA and its mutants, the reconstructed pRSFDuet-M2-ComA plasmid was transformed and expressed in BL21(DE3) *E. coli* cells. 0.5 mM Isopropyl β-D-1-thiogalactopyranoside (IPTG) was added when cells $OD_{600}$ reached to 0.8 and cultured for another 20 h at 18°C. Harvested cell pallets were resuspended using Buffer A (50 mM Tris-HCl, 300 mM NaCl, 10%glycerol, pH 7.5) in the presence of 1 mM dithiothreitol (DTT) and 1 mM phenylmethylsulfonyl fluoride (PMSF). Cell suspension was broken by a high-pressure homogenizer and ultracentrifuged to get the cell membranes. Buffer A with 2% LMNG (Anatrace) was used to solubilize membranes for 2 h at 4°C, and then the supernatant was subjected to further purification with anti-MBP beads (SMART lifesciences). Buffer A in the presence of 60 mM maltose and 0.01% LMNG was applied to elute ComA proteins. MBP tag was cleaved by tobacco etch virus protease (TEVP) for further size-exclusion chromatography (SEC) purification with Superose 6 Increase 10/300 GL column (Cytiva). Fractions corresponding to ComA were pulled and concentrated to 4.5 mg/ml for the following studies.

For the truncated PEP domain, the reconstructed PRSFDuet-M2-PEP plasmid was transformed and expressed in BL21(DE3) *E. coli* cells. 0.5 mM Isopropyl β-D-1-thiogalactopyranoside (IPTG) was added when cells $OD_{600}$ reached to 0.8 and cultured for another 15 h at 18°C. Cells were collected and resuspended using Buffer B (50 mM Tris-HCl, 100 mM NaCl, 2%glycerol, pH 7.5) with 1 mM DTT 1 mM PMSF. Cell suspension was subsequently broken by a high-pressure homogenizer and cell debris was moved using a high-speed centrifuge. The supernatant was collected and applied to MBP affinity purification and

eluted with Buffer B in the presence of 60 mM maltose. The MBP-tag was moved by incubation with 1 mg/ml TEVP at 30 °C for 30 min, and concentrated for further purification by SEC with Superdex 200 Increase 10/300 GL column. Fractions corresponding to PEP domain were pulled and concentrated for the peptidase assay.

For ComC and its mutants, the corresponding pET-15b-ComC plasmids were transformed in *E. coli* DE3 and cultured to $OD_{600} = 0.3$–0.4 for expression induced with 0.5 mM IPTG for 1 h at 37 °C. Harvested cell pellets were resuspended with Buffer A and were broken using a high-pressure homogenizer. Supernatant from the high-speed centrifuge was further applied to His-affinity purification. Buffer B in the presence of 300 mM imidazole was used to elute the ComC protein and concentrated for size-exclusion chromatography with Superdex 200 Increase 10/300 GL column (Cytiva). The concentration of purified ComC was determined by microvolume UV-Vis Spectrophotometer (Thermo Fisher) at 257 nm, based on the extinction coefficient of phenylalanine[53]. Tricine-SDS-PAGE[54] was used for ComC characterization.

### Peptidisc reconstitution
Peptidisc reconstitution was performed as previously described[34,55]. In short, ComA was solubilized using 2% n-dodecyl-β-D-maltoside (DDM) (Anatrace), followed by MBP affinity purification. After binding to the beads, the protein was washed with ten-column volumes (CV) of Buffer B in the presence of 1 mg/ml peptidisc peptide (Peptidisc Biotech), and then eluted in Buffer C (50 mM Tris-HCl, pH 7.5, 100 mM NaCl, 60 mM maltose). MBP tag was cleaved by TEVP for further SEC purification with Superose 6 Increase 10/300 GL column (Cytiva). Purified peptidisc with ComA reconstituted was further concentrated for functional study.

### Peptidase activity assay
Purified ComC substrate was incubated with ComA enzymes at a molar ratio of 15:1 for indicated time and temperature. Total volume of the reaction system was 10 μL. ComC cleavage was analyzed by 12% Tricine-SDS-PAGE gel and Image J software[56] for intensities of protein bands. To test the effect of nucleotide on the peptidase activity, ATP, ATP analogs including ATPγS, AMP-PNP, and Vanadate, $MgCl_2$ and EDTA were added at a final concentration around 3 mM.

### Nanodisc reconstitution of ComA
The membrane scaffold protein MSP1D1 was expressed in *E. coli* BL21(DE3) and subsequently purified via Nickel-Nitrilotriacetic acid (Ni-NTA) and Size Exclusion Chromatography (SEC) using a Superdex 200 Increase 10/300 GL column (Cytiva) in a buffer containing 50 mM Tris-HCl (pH 7.5) and 100 mM NaCl. The MSP1D1 was then concentrated to a final concentration of 5 mg/ml. *E. coli* total lipid was prepared by solubilization to a concentration of 10 mg/ml using 200 mM sodium cholate. In parallel, ComA with a cleaved MBP tag, suspended in 0.02% DDM, was concentrated to 2 mg/ml.

The nanodisc reconstitution procedure employed in this study closely follows the previously described method[57]. Specifically, for ComA reconstitution, the molar ratio used was ComA: MSP1D1: E. coli total lipid=1:2:150. The final sodium cholate concentration in the mixture was set to 25 mM. Bio-beads SM2 (Bio-Rad) were prepared at a ratio of 100 mg per 1 ml of the mixture, added to initiate the reconstitution process, which works by extracting detergents from the system. After an hour of incubation at 4°C with constant rotation, the mixture was left to incubate overnight under the same conditions. Following the incubation period, the Bio-beads were removed, and the reconstituted sample was clarified via centrifugation. The sample then underwent separation on an SEC Superose 6 Increase 10/300 GL column in the same buffer used previously (50 mM Tris-HCl, pH 7.5, 100 mM NaCl). The resulting samples were analyzed by SDS-PAGE and subsequently concentrated for further use.

### ATPase activity assay
All ATPase activity assays were performed using a previously described procedure[58] with minor modifications. First, 2 μg of purified ComA or ComA mutants was incubated with 50 μl reaction solution of 50 mM HEPES, pH 7.5, 10% glycerol, 100 mM NaCl, 2.5 mM ATP and 5 mM $MgCl_2$, and incubated in a water bath at 37 °C for 30 min. The reaction was stopped by adding 50 μl 12% (w/v) SDS. Then the reaction was incubated at room temperature for 5 min after the addition of 100 μl solution containing 12% (w/v) ascorbic acid in 1 M HCl and 2% (w/v) ammonium molybdate in 1 M HCl. Finally, 150 μl solution containing 25 mM sodium citrate, 2% (w/v) sodium metaarsenite, and 2% (v/v) acetic acid was added, and incubated at room temperature for 10 min. Absorbance at 848 nm was measured using a multimodal microplate reader (HIDEX), and a standard curve of potassium phosphate ranging from 0.05 mM to 0.6 mM was generated to quantitate the amount of released phosphate. Reaction reagents were purchased from Sigma.

### Electron microscopy sample preparation and data acquisition
3 μl of purified ComA or ComA mutants in detergents at a concentration of 4.5 mg ml/ml was applied to glow-discharged Quantifoil holey carbon grids (1.2/1.3, 400 mesh). For the ATP complex, a final concentration of 20 μM ComC and 2 mM ATP were added into ComA (E647Q) and incubated at room temperature for 15 min before freezing. For the CSP complex, final concentrations of 20 μM ComC were added and incubated at 25 °C for 25 min before applying the mixture to cryo-EM grids. For the ATPγS complex, final concentrations of 20 μM ComC and 2 mM ATPγS/$Mg^{2+}$ were added and incubated at 37 °C for 25 min before applying the mixture to cryo-EM grids. For the ATP/$Zn^{2+}$ complex, final concentrations of 2 mM ATP/$Zn^{2+}$ were added and incubated at 37 °C for 25 min before applying the mixture to cryo-EM grids. Grids were blotted for 3–4.5 s with 100% relative humidity and plunge-frozen in liquid ethane cooled by liquid nitrogen using a Vitrobot System (Gatan). Cryo-EM data were collected at liquid nitrogen temperature on a Titan Krios electron microscope (Thermo Fisher Scientific), equipped with a K3 Summit direct electron detector (Gatan) and GIF Quantum energy filter. All cryo-EM movies were recorded in counting mode with SerialEM4[59] with a slit width of 20 eV from the energy filter. Movies were acquired at a nominal magnification of 81,000×, corresponding to a calibrated pixel size of 0.858 Å on the specimen level. The dose rate was set to be 7.6 counts per physical pixel per second. The total exposure time of each movie was 6 s, resulting in a total dose of 46.4 electrons per Å², fractionated into 40 frames (150 ms per frame). More details of electron microscopy data collection parameters are listed in Supplementary Table 2.

### Electron microscope image processing
EM data were processed as previously described[42]. Dose-fractionated movies collected using K3 Summit direct electron detector were subjected to motion correction using the program MotionCor2[60]. A sum of all frames of each movie was calculated following a dose-weighting scheme, and used for all image-processing steps except defocus determination. CTFFIND4[61] was used to calculate defocus values of the summed images from all movie frames without dose weighting. Particle picking was performed using a semi-automated procedure with SAMUEL and SamViewer[62].

For particle picking, we used 2D averages from our prior FtsEX study as templates[63], which included five different side views. We increased the contrast of the motion-corrected images by binning them four times, then applied a 48-pixel box for particle picking. An initial set of around 200 images was used for preliminary particle selection, followed by a basic 2D classification using SAMUEL[64]. The five most accurate 2D averages, exhibiting intact ABC transporter features, were chosen for another round of template-based particle picking on all motion-corrected images. The picked particles underwent screening through a 2D classification using SAMUEL. We selected

particles from the 2D averages displaying clear ABC transporter features, specifically a rectangular-shaped TMD region. Averages showing two distinct dots at the center, corresponding to the top or bottom views of the sample, were also chosen, although side views were sufficient for subsequent 2D and 3D reconstructions.

Post-SAMUEL screening, selected particles were extracted from the dose-weighted, unbinned, motion-corrected images using a 256-pixel box size. The subsequent 2D and 3D classifications and 3D refinement were conducted using "relion_refine_mpi" in RELION3[65]. At each step, we only retained particles from the group displaying complete ABC transporter features and relatively high resolution compared to other classes in the results for further processing. All refinements followed the gold-standard procedure, in which two-half data sets were refined independently. The overall resolutions were estimated based on the gold-standard criterion of Fourier shell correlation (FSC) = 0.143. Local resolution variations were estimated from two half-data maps using ResMap[66]. The amplitude information of the final maps was corrected by "relion_post_process" using the program RELION3[65].

EM data of ATP/$Zn^{2+}$ ComA were processed by CryoSPARC[67]. Dose-fractionated movies collected using K3 Summit direct electron detector were subjected to motion correction using the program MotionCor2[60]. A sum of all frames of each movie was calculated following a dose-weighting scheme and used for all image-processing steps except defocus determination. CTFFIND4[61] was used to calculate defocus values of the summed images from all movie frames without dose weighting. Particle picking was performed using the blob picker followed by the template picker. 2D and 3D classification and 3D refinement were carried out using "2D classification", "Ab-initial Reconstruction" and "Heterogenous Refinement". Refinements were done using "Homogenous Refinement" and "Non-Uniform Refinement". The overall resolutions were estimated based on the gold-standard criterion of Fourier shell correlation (FSC) = 0.143. Local resolution was estimated by "Local Resolution Estimation".

## Model building and refinement

The initial model of ComA was generated by SWISS-MODEL server[68] using the crystal structure of PCAT1 (PDB ID: 4S0F) as the template. This initial model was rigid-body fitted to our cryo-EM maps in UCSF Chimera[69], extensively rebuilt in Coot[70], and refined using real space refinement in Phenix[71]. Restraints for ATP and ATPγS were generated with phenix_elbow program using its isomeric SMILES string files obtained from the PDB Chemical Component Dictionary through Ligand Expo. Ligands were manually docked into the cryo-EM maps in Coot, followed by iterative real-space refinements in Phenix. Final models were validated with statistics from Ramachandran plots, MolProbity scores, and Clash scores with the program in Phenix (see Supplementary Table 2 for details). Figures were generated using UCSF Chimera.

## comC-HiBiT secretion assay

Primers used for the synthesis of ComC-HiBiT expressing strains are listed in Supplementary Table 4. Genetic transformations of *S. pneumoniae* were performed as described previously[72]. Briefly, Pneumococcal cells were cultured in brain heart infusion broth (BHI) (Thermofisher Scientific) at 37°C in 5% $CO_2$. PCR products were synthesized using high fidelity Phusion DNA polymerase (NEB M0530S) and purified with the QIAquick PCR purification kit (Qiagen 28106) following the manufacturer's protocol. Cells were transformed with cassettes assembled by isothermal assembly after the induction of natural competence. Transformants were selected on blood plates supplemented with the indicated antibiotics. Allelic replacements were performed using the Janus cassette (P-*spec-rpsL*)[73]. The resulting strains were validated by diagnostic PCR using GoTaq DNA polymerase (Promega, M712) and Sanger sequencing. Antibiotics were purchased

from Sigma-Aldrich and used at final concentrations of 0.3 μg/ml for erythromycin (Erm), 150 μg/ml for spectinomycin (Spec), and 300 μg/ml for streptomycin (Str).

Mutants of ComA and ComC expressing ComC-HiBiT were generated with primers listed in Supplementary Table 4, and grown in BHI broth and incubated at 37°C in 5% $CO_2$. To prevent the secretion of CSP by the BlpAB transporter[52], strains expressing ComC-HiBiT are in a Δ*blpA* background. When the optical density at 600 nm ($OD_{600}$) reached 0.1 to 0.3, cultures were normalized to $OD_{600}$ of 0.1, and exogenous CSP was added to induce natural competence. After an hour of induction, cultures were immediately placed on ice for 5 minutes. Cells were pelleted by centrifugation at 16,100x *g* for 2 minutes at 4°C. The supernatant fraction was collected and stored at 4°C. Pellets were washed with prechilled BHI twice and the washed pellets were then resuspended in 1 ml of prechilled BHI. The suspension was transferred to the lysis matric column homogenizer and disrupted 10X for 3 rounds of lysis. Each round consists of 3 cycles at 6 M/s for 40 seconds. Suspensions were placed on ice for 5 minutes after every round of homogenization. Cell suspensions were centrifuged at 16,100x *g* for 2 minutes at 4 °C to remove cellular debris. The Δ*comA* and Δ*comC* strains were used as negative controls. Supernatant and cell lysate samples of the ComC-HiBiT-expressing strains were aliquoted into a white 96-well plate (Corning). The HiBiT tag in the samples were quantified by adding an equal volume of the HiBiT Extracellular Detection Reagent (Promega, N2421) to each well. Luminescence was measured using a Tecan plate reader with the following settings: 1 s integration time, gain 135. Experiments were repeated three times, and the differences between mutants were evaluated by the Mann-Whitney U test.

## Cross-linking mass spectrometry and data analysis

Purified ComA proteins were cross-linked in buffer A for pre-cleavage and buffer B for post-cleavage state with the addition of 1 mM disuccinimidyl sulfoxide (DSSO, Thermo A33545), with 26:1 ratio for DSSO:protein by molarity in DMSO (Final 10% DMSO concentration) for 60 min at 25 °C with shaking. Cross-linking was then quenched by adding 1 M Tris to the final concentration 80 μM. Cross-linked protein was then run into SDS-PAGE gel and the gel was stained with premixed colloidal coomassie G-250 staining solution (BioRad 1610786). Bands with molecular weight corresponding to cross-linked ComA were excised and proceeded with in-gel digestion. Briefly, gel bands were incubated three times with 50% ethanol in 100 mM TEAB, triethylammonium bicarbonate, with gentle agitation for 5 min. Then the gel was rehydrated with 100 mM TEAB followed by dehydration with 100% ethanol which was performed successively for two times. Excess ethanol was removed by short vacuum centrifugation. The gel pieces were then rehydrated with 20 mM TCEP, tris(2-carboxyethyl) phosphine, in 100 mM TEAB and incubated at 55 °C for 60 min and left to cool to room temperature. Next, 500 mM CAA, chloroacetamide, was added to final 55 mM concentration and incubated in the dark for 30 min. The gel was then washed twice with 100 mM TEAB and dried by vacuum centrifugation. 2 gel replicates were then digested by addition of trypsin (Pierce 90058), 2 μg in 100 mM TEAB, and 2 gel replicates were digested by addition of chymotrypsin (Promega V1061), 2 μg in 100 mM TEAB, 10 mM CaCl2, and incubated at 37 °C for 16 h. Protease digestion was quenched by the addition of TFA, trifluoroacetic acid, to a final concentration of 1% (v/v). Peptides were extracted by the addition of 30% ACN, acetonitrile, followed by 100% ACN. Extracted peptides were pooled and dried by vacuum centrifugation. Dried peptides were then resuspended in water with 0.1% formic acid and desalted with C18 stage tips (Empore C18 discs). Briefly, stage tips were activated with 100% ACN then equilibrated twice with water with 0.1% formic acid. Cross-linked peptides were loaded and the stage tip was washed twice with water with 0.1% formic acid. Peptides were eluted with 65% ACN, 0.1% formic acid and dried by vacuum centrifugation.

Liquid chromatography-mass spectrometry (LC-MS) acquisition of cross-linked peptides was performed as previously described[74]. Desalted cross-linked peptides were resuspended in 0.5% acetic acid, 0.06% TFA, 2% acetonitrile in water. 1 μg of cross-linked peptide was injected on a Easy-nLC 1200 (Thermo) chromatography system coupled to a Orbitrap Fusion Lumos mass spectrometer (Thermo) using a 50 cm × 75 μm inner diameter Easy-Spray reverse phase column (C-18, 2 μm particles, Thermo) over a 60 min gradient from 0.1% formic acid in water to 40% acetonitrile with 0.1% formic acid. MS acquisition was performed with MS1 using Orbitrap 60 K resolution with scan range 350–1650 m/z. Precursor ions with 3–8 positive charge were selected for MS2 CID fragmentation with normalized collision energy of 30% and Orbitrap analyzer at 30 K resolution. MS3 HCD fragmentation was triggered based on targeted mass difference of DSSO (31.9721 Da) for 4 dependent scans with normalized collision energy of 30% and Ion Trap analyzer in rapid mode.

Thermo raw files were searched against fasta file containing ComA protein sequence using Metamorpheus v0.0.318 with a calibration search with precursor mass tolerance of 10ppm and product mass tolerance of 20ppm. Cross-link search was performed for DSSO on K, S, T, Y amino acids for MS2 CID and MS3 HCD scans with 3 maximum missed cleavages, trypsin protease and fixed modification for carbamidomethyl (C) and variable modifications for oxidation (M), deamidation (N,Q) DSSO hydrolyzed by water and hydrolyzed by Tris (protein N-terminus, K, S, T, Y), DSSO alkene and thiol (protein N-terminus, K, S, T, Y). Crosslinks from intralinks result files were filtered for $q$ value ≤ 0.01. Cross-links were visualized by xiVIEW. Raw mass spectrometry spectra and search data were uploaded to the jPost repository[75].

### Reporting summary

Further information on research design is available in the Nature Portfolio Reporting Summary linked to this article.

## Data availability

The data that support this study are available from the corresponding author upon request. The cryo-EM density maps of ComA in the presence and absence of bound CSP, nucleotides or divalent metal have been deposited in the Electron Microscopy Data Bank under accession codes: EMD-34712 (ATPγS-bound ComA, OF-state with EC gate closed); EMD-34713 (ATPγS-bound ComA, OF-state with EC gate open); EMD-34714 (ATP-bound ComA (E647Q), post-cleavage state); EMD-34715 (CSP-bound ComA); EMD-34716 (ComA C17A with ComC), EMD-36882 (ATP-$Zn^{2+}$ bound ComA), EMD-36936 (ATP-$Mg^{2+}$ bound ComA (E647Q)). Atomic models have been deposited in the Protein Data Bank under accession codes 8HF4 (ATPγS-bound ComA, OF-state with EC gate closed), 8HF5 (ATPγS-bound ComA, OF-state with EC gate open), 8HF6 (ATP-bound ComA (E647Q), post-cleavage state), 8HF7 (CSP-bound ComA), 8K4B (ATP-$Zn^{2+}$ bound ComA) and 8K7A (ATP-$Mg^{2+}$ bound ComA (E647Q)). Raw mass spectrometry spectra and search data were uploaded to the jPost repository with the following accession numbers: JPST001916 (jPOST) and PXD038058 (ProteomeXchange). The source data underlying Figs. 1b, 1c, 1d, 1e, 1f, 4c, 4e, 4f, 6d, 7a-d and Supplementary Figure 1a, 1b, 1c, 1d, 1e, 1f, 6c, 9, 18 are provided as a Source Data file. Source data are provided with this paper.

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

## Acknowledgements

We are grateful to Adam Yuan at the Department of Biological Sciences, National University of Singapore for the reagents and equipment. We thank the staff at cryo-EM Center of National University of Singapore, Institute of Physics at China, and Harvard Medical School in the United States, for their help in initial sample screening and final data collection. We thank Tom Walz at The Rockefeller University for helpful discussion and comments on the project. This work is supported by grants from the National University of Singapore Start-up grant to M. L., and the National Research Foundation Fellowship (NRF) (NRFF11-2019-0005 to L.T.S.), the MOE Tier 2 Fund (MOE-T2EP30222-0015 to M. L., MOE-T2EP30220-0012 to L.T.S.). The CLMS-related research was supported by A*STAR core funding to Z.S. and R.M.S. R.M.S is also supported by Singapore National Research Foundation under its NRF-SIS (NRF2017_SISFP08) "SingMass" share infrastructure scheme. Z.S. is also supported by an A*STAR Career Development Fund (212D800074) and by the A*STAR Young Achiever Award.

## Author contributions

M.L. conceived the project. L.Y. performed all biochemical reconstitution and functional characterization in vitro; X.X. performed EM data collection and model building; W.Z.C. performed in-cell studies; H.F. performed the Zn2+ inhibition assay and determined the Zn2+-inhibited structure of ComA; K.S. helped in model building; S.Z. and R.S. performed CLMS analysis; J.S. helped with EM data collection; Y.M.W. and Z.L.L. helped with initial sample screening; L.T.S. designed and supervised the in-cell studies and helped in the writing; M.L. designed and supervised research, analysed data, and wrote the paper with help from all authors.

## Competing interests

The authors declare no competing interests.
