## [Peer Review File · Nature Communications]

Structural basis of peptide secretion for Quorum sensing by ComAReviewers' Comments:

Reviewer #1:

Remarks to the Author:

The manuscript by You et al., describe the characterization and structural elucidation of the PCAT transporter ComA that is involved in proteolysis ComC and subsequent secretion of the competency stimulating peptide involved in quorum sensing. Based largely on biochemical studies derived from five cryoelectron microscopic studies, the authors propose a role for divalent metal ions in triggering the outward facing conformation of the transporter. The work may have significant implications in our understanding of PCAT systems but several issues must be addressed before the current manuscript may be considered for publication in a high-profile journal like Nature Comm.

1. A major limitation of the current study is that none of the structures visualize the C39 protease domain. Hence, there is a lack of information on how the leader peptide is recognized by the transporter prior to extrusion into the translocation cavity. The authors gloss over this omission very quickly and delve into fine details of what structures they are able to produce on hand. This lack of information on the PEP domains must be clearly stated in the text, and especially in the legend to Figure 6.
2. The authors use the term OF-open and OF-closed to suggest changes in the orientation of the TMD before and after cargo release. This terminology is too confusing and should be substituted with more meaningful terms so that the casual reader does not confuse the OF-closed term with the pre-catalytic closed conformation.
3. Prior studies on other PCAT systems (i.e. PCAT1) suggest that release of peptide cargo causes a change to an occluded state. Can the authors reconcile their model in Figure 6 with conformational models presented in the literature for other similar systems?

In short, the work is done nicely and there are certainly new advances here. However, the authors over-interpret some of their observations and do not do a service of accounting for prior studies done by others on similar systems. A revised version that addresses these concerns should be suitable for Nature Comm.

Reviewer #2:

Remarks to the Author:

The manuscript by Yu et al. explores the structural basis of competence-stimulating peptide (CSP) secretion by ComA, which plays a key role in bacterial quorum sensing. Authors reconstituted ComA in peptidiscs and demonstrated ComC cleavage was inhibited by the presence of ATP, similar to other PCAT family members. Using single particle cryo-EM, the authors solved the structure of ComA in several conditions: (1) ComA (E647Q)+ComC+ATP, (2) ComA+ComC, (3) ComA+ComC+ATPyS-Mg²⁺, (4) ComA+ATP-Zn²⁺. Authors identify a possible CSP binding pocket in their structure and note local rearrangements of the EC gate that seemed to depend on ATP-Mg²⁺ binding to the NBDs. Based on their structures, authors mutated key residues in the ComA cavity and show their effects on protease activity and a cell-based CSP secretion assay. The structure of ComA and identification of a potential CSP binding pocket are key findings that may potentially be of interest in the peptidase-containing ABC transporter field.

However, we have a number of significant concerns about this work in its current form. While the cryo-EM structure of *Streptococcus pneumoniae* ComA is novel, the modeling of the cryoEM maps is not well executed and the interpretation of the structural data lacks rigor, particularly in areas where the authors make key claims in the manuscript such as the putative CSP binding site in ComA and the ATP-binding site in the NBD. What inferences can be drawn for the biochemical assays is also unclear

to us, and while mutations are made that alter peptidase activity, the mechanistic basis is largely unexplored, even when the results are somewhat surprising. More broadly, the authors make strong claims throughout the manuscript despite having weak evidence supporting those claims. This manuscript will require substantial revision, which may change the key claims made.

MAJOR COMMENTS:

(1) In Lines 104-108: While a peptidisc may yield native-like activity for some membrane proteins, I think it cannot be assumed to be true always, and must be demonstrated for each protein to be studied. Unlike a nanodisc, in a peptidisc there is no added lipid added around the membrane protein; it is effectively solubilized by the belt protein any any residual detergent or lipids that aren't removed in the exchange. Many ABC transporters exhibit substrate stimulated ATPase activity, but ComA does not in LMNG or peptidisc, which may indicate that it needs lipids to recapitulate its native mechanism (it is also possible that it just isn't simulated by substrate). We think the authors should be more conservative in their wording here. Also, Fig 1B seems to indicate a comparison between activity in "Detergent" vs "Nanodisc", but I think this is incorrect and that only peptidiscs were used here.

(2) In Lines 139-147; Table S1: It seems like there are generally fewer crosslinked peptides detected in the ATP-bound sample versus apo. Might the crosslinks not observed in the ATP-bound sample be simply missed due to lower coverage? Can the data be analyzed in a more rigorous way? How many replicates were performed? No stats are reported indicating whether the differences are significant, and no indication of peptide abundance is presented.

(3) Figures often show error bars, but the legends don't state what they represent. They also don't state the number of replicates carried out, and whether the replicates were technical or biological.

(4) The authors model protein side chains and ligands in regions of their cryo-EM map that are lower resolution and contain weaker densities. This is problematic since the authors make claims in the manuscript related to these regions such as the CSP pocket and ATP-binding site, which may be an overinterpretation of their experimental maps. In several figures (eg. Fig. 3A, 4C, 4D, 4G), authors show cryo-EM densities carved around poorly modeled ligands which is deceptive since it masks the surrounding protein and makes the ligand appear to have stronger density.

For example, in Fig. 3A, the actual map doesn't look like this figure. The map contour would have to be unreasonably low and then carved to make a figure like this. This is misleading and how the map was carved or modified to make this figure should be stated clearly in the legend. In general, it is helpful to include information about contour level and how the density was generated (volume zoning radius, etc.) for any figure containing cryo-EM density.

(5) In their CSP-bound-ATP-free map, two CSPs are modeled in the putative binding pocket in ComA. Since the densities for the two ligands are generally weak, a major question is "what is it?" and what would be an appropriate way to model it? For starters, this sample has probably seen a whole lot of other molecules along the way, including lipids and detergent. It would be very difficult for the authors know that this is the peptide and not something else, and no other evidence presented in this work directly supports that the peptide binds this site. Second, even if this were density for the peptide (admittedly, it might be), in our opinion, this map does not allow the assignment of the direction of the peptide (N- vs C- termini) nor the sequence register. This is evident in the relatively poor map-model scores for chain C (0.5146) and chain D (0.5049) and relatively high clashcore (15) of the model. So at best, I think the authors could say they observe density at this site, which may be a peptide but potentially could be something else. In the deposited model, this ambiguity should be reflected by labeling anything explicitly modeled as UNK or UNL. Orthogonal evidence such as cross-linking mass spectrometry on a sample of ComA+ComC or binding assays between ComA+ComC could be useful in validating the interaction between CSP and ComA and whether this pocket is specific for this peptide.

(6) In their reconstruction of the CSP-bound-ATP-free map (Fig. S6), authors applied C2 symmetry whereas C1 is used for all other other reconstructions in this study. What is the authors' reasoning for applying symmetry? Previous structures of related transporters such as PCAT1 (PDB 6V9Z, Kieuvongngam et al., eLife, 2020) show that the peptide cargo binds asymmetrically to the transporter. Application of C2 symmetry may present artifacts in the reconstruction, particularly in the region where CSP binds. I would suggest that the authors try to improve the resolution of this region by performing 3D classification and focus refinement in C1. For example, 3D classification may reveal that only 1 CSP molecule is present and that it can occupy both halves of the binding pocket equally.

(7) In Lines 229-231: I would expected the opposite, that peptide bound to the TMD would inhibit PEP activity, but mutation of the putative binding site (presumed to reduce peptide binding, but not demonstrated) actually inhibits PEP activity. The authors don't provide a rationale for their observations. More broadly, the absence of ATPase stimulation by substrate complicates the authors biochemical analysis, leading them to use peptidase activity as a readout instead. However, why substrate binding to the TMDs should affect peptidase activity is not clear, thus exactly what information the peptidase assay provides is not clear to us. The basis for this assay and the logic is unclear to me. Why is PEP activity a good read out for mutations in the putative peptide binding site? It seems like a binding assay would be more appropriate.

(8) In Lines 296-306: I would say this analysis is not well supported by the data. Just looking at Fig. 4, C-D, there not great density for the surrounding side chains even, let alone a map that would allow one to determine whether Mg is bound or not. And while the authors are trying to interpret if individual atoms are present, one of the ATP-gamma-S molecules build into the OF-open structure is mis-fit in the map such that the base is rotated ~ 180 degrees from how it is normally oriented in ABC transporters (Chain A, 801), though the map clearly indicates that this is incorrectly modeled. The other ATP-gamma-S doesn't have this issue, but instead the sulfur of ATP-gamma-S is only ~ 1.3 Å from the Mg, which is physically impossible. It would seem that the data is not sufficient for the authors to place even groups as large as adenosine accurately in these maps, so I question their ability to assess whether Mg is present or absent.

(9) Based on State 2 that was observed in their ComA+ComC+ATPyS-Mg²⁺ sample, authors claim that ATP-Mg²⁺ complex is needed to open the EC gate and release CSP. However, closer inspection of the cryoEM map of State 2 reveals that densities globally similar to those observed in the "CSP-bound" structure are still present, perhaps only slightly weaker. Claims that this structure has released CSP are therefore not on solid ground.

(10) In Lines 308-321: I think the rationale behind this experiment is not clear, nor are the biological implications. It is known that ABC transporters and many other ABC transporters use Mg, and occasionally can tolerate other divalents like Mn. It is known that Zn usually doesn't work well. Apart from demonstrating that this ABC transporter also doesn't use Zn very well, it's not clear to me what this adds beyond the OF-open structure in the presence of ATP-gamma-S. Similarly, given Mg is abundant in the cell, under what circumstances would Mg not bind almost simultaneously with the ATP? This seems to be a major point of the paper, and comes back again in the discussion (In 387-393), but I think ATP and Mg may actually be pre-complexed in the cytoplasm and bind the transporter simultaneously. In the cell, there likely is never an ATP-only state, so the relevance of this seems very limited.

(11) The resolution of the uploaded ATP-Zn ComA map seems much lower than what is shown/reported in Fig. S12 (3.8 Å-resolution). The uploaded map has very little side chain information, yet the authors have model all the residues of ComA into the map with an overall clashscore of 51. Is the reported resolution a typo? Sent us the wrong map as well as refined the model with sub-optimal parameters? Additionally, the authors have modeled in ATP-Zn into the map despite having poor resolution for the substrate as well as the surrounding protein. Since a lot of the protein is unaccounted, density that is zoned around the ATP-Zn in Figure 4G may also include

unmodelled protein.

(12) In Lines 332-342: The interpretation of these experiments is confounded by the fact that the authors show in Fig. 3 that many of these mutants modulate PEP activity, and it seems like their impact of PEP activity generally mirrors their impact on "transport". What is being measured here seems to be a combination of peptide cleavage and transport, yet it seems like the results could potentially be explained by the processing defect from Fig 3 with no change in transporter activity. Is transport actually affected at all by these mutants? Can the peptide be provided to the transporter in a way that uncouples the peptidase function from the transport function, perhaps by co-expressing soluble PEP domain and ComC, along with ComA with a deletion or mutation of the PEP domain?

MINOR COMMENTS:

(1) In 29-30: "determine five cryo-EM structures in the presence or absence of CSP or ATP": Strictly speaking, this is incorrect, as 2 have ATP-gamma-S bound; similar issue In 89-90

(2) In 103: Perhaps "solubilized" would be a better word than "dissolved"?

(3) Fig. S1D: Is there an error in the labels above the gel? The first lane after the ladder has no labels, but is perhaps ComC alone? But then the next lane, which is indicated to be ComC alone, appears to be fully processed ComC cleaved to CSP and leader. I am therefore not sure what to conclude from the other lanes.

(4) Figure S2. Would it be possible to color code the cross-links to better highlight the X-links that are unique to the ATP-free sample versus ATP bound sample.

(5) In 130-131: These results are interesting, but I am having trouble understanding how to interpret this result and the authors merely present the data. Isn't it a bit surprising that ATP vs non-hydrolyzable analogs have the same effect? I might have at least expected ATP+Mg to have an intermediate effect, since it can turn over while the other might be trapped in a particular conformation. Strictly speaking, the authors don't show the effect of Mg-along or ATP-alone, so perhaps one can't rule out the possibility that Mg is inhibiting cleavage somehow? It is also unclear at this point if inhibition is mediated by ATP/analog binding to the NBDs of the ABC transporter, or somewhere else. A mutation that blocks ATP binding to the NBDs would provide evidence that inhibition is mediated via the NBDs.

(6) In 160. 0.5 mM ATP written here but 2 mM ATP was used according to Methods (Line 495).

(7) In 185-192: The experiment and logic are not well described here. What is being mutated? How is making multiple mutations and destabilizing the protein contrary to the functional studies?

(8) In 194-195: Perhaps some comparison to past work would make sense here, as this was probably expected based on the structures of PCAT (PMID: 26201595) and perhaps also IrtAB (PMID: 32296173), rv1819c (PMID: 32296172), YbtPQ (PMID: 32076651), etc. Also, it seems like a structure of the ComA NBDs from another species was reported, but no comparison was made to this structure.

(9) In 203-206: This is confusing. If in the next section, evidence will be presented that CSP does not bind here, why speculate that it may bind here, but that it probably doesn't?

(10) Figure 3B: Color hydrogen bonds vs salt bridges different colors

(11) Figure 3D: Colors of labels are switched. K labels should be blue

(12) In 269: No data for CSP release is shown. Section title should be reworded to not be so speculative.

(13) In 297: I suggest deleting "high atomic resolution", as 3 Å is still very far from atomic resolution. This is clearly illustrated by my comment on In 296-306.

(14) It seems that the PEP domain is never visible in their maps, yet somehow the conformation of the transporter/nucleotide state is regulating the peptidase activity. How do the authors imagine this is happening?

(15) In 514-539. Steps for cryoEM image processing are sparse. It would be great if authors could include more details for readers about how they processed their data (e.g. criteria for excluding certain classes during 3D classification, when masks were applied, etc.)

Dear Reviewer,

Thank you for your positive feedback and valuable insights. A point-to-point reply addressing your comments is provided below. The responses are labeled in blue for clarity, and line numbers are provided based on the **marked** file. We appreciate your time and the opportunity to improve our work.

REVIEWER COMMENTS

Reviewer #1 (Remarks to the Author):

The manuscript by Yu et al., describe the characterization and structural elucidation of the PCAT transporter ComA that is involved in proteolysis ComC and subsequent secretion of the competency stimulating peptide involved in quorum sensing. Based largely on biochemical studies derived from five cryoelectron microscopic studies, the authors propose a role for divalent metal ions in triggering the outward facing conformation of the transporter. The work may have significant implications in our understanding of PCAT systems but several issues must be addressed before the current manuscript may be considered for publication in a high-profile journal like Nature Comm.

We thank the reviewer for the very positive comments. The issues raised by the reviewer have been addressed. We found the revised manuscript much improved compared to the original version.

1. A major limitation of the current study is that none of the structures visualize the C39 protease domain. Hence, there is a lack of information on how the leader peptide is recognized by the transporter prior to extrusion into the translocation cavity. The authors gloss over this omission very quickly and delve into fine details of what structures they are able to produce on hand. This lack of information on the PEP domains must be clearly stated in the text, and especially in the legend to Figure 6.

Fully agreed, and we have included a new cryo-EM map of the ComA (C17A) mutant resolved from a sample in the presence of the ComC peptide substrate (**L342-L365, Fig. 2**). While the resolution is somewhat limited (~6 Å), the electron densities corresponding to the two PEP domains are now clearly visualized.

Particularly, this map illustrates that in the presence of ComC, the two PEP domains are ordered and associated to the main body of the ABC transporter, positioned precisely at the two intracellular (IC) gate sites that link to the central cavity of the transmembrane domain (TMD). This positioning facilitates the subsequent entry of CSP into the translocation channel across the membrane.

Furthermore, while the PEP domains are connected to the ABC transporter, the two nucleotide-binding domains (NBDs) are widely separated from each other. Despite the lack of high-resolution data that would allow for a detailed dissection of leader peptide binding and cleavage, these observations align perfectly with those reported in the PCAT1 study (PMID: 26201595, 31934861).

The new structural insights obtained from the PEP domain, together with our functional analyses and cross-linking mass spectrometry (CL-MS) investigations into CSP processing (**L317-L461, Fig. 2**), lay a robust groundwork for our suggestion that peptide processing is probably conserved across the PCAT family. As recommended, we have integrated the elegant peptide processing study from the PCAT1 (PMID: 26201595, 31934861) into our working model (now updated to **Fig. 8** in the revised version), and have ensured this source is clearly acknowledged.

2. The authors use the term OF-open and OF-closed to suggest changes in the orientation of the TMD before and after cargo release. This terminology is too confusing and should be substituted with more meaningful terms so that the casual reader does not confuse the OF-closed term with the pre-catalytic closed conformation.

Agreed. "OF-open" has been rephrased to "OF-state with the EC gate open", while "OF-closed" has been altered to "OF-state with the EC gate closed". This should more accurately reflect the comparison with the "OF-state with the EC gate open" resolved in the same sample, and we hope this clarification will help eliminate any confusion.

3. Prior studies on other PCAT systems (i.e. PCAT1) suggest that release of peptide cargo causes a change to an occluded state. Can the authors reconcile their model in Figure 6 with conformational models presented in the literature for other similar systems?

Our study has yielded two EM structures that exhibit a significantly occluded central cavity within the transmembrane domain (TMD) upon nucleotide binding: 1) the E647Q mutant with ATP bound, and 2) the 'state 1' conformation identified in our research in the presence of ATPγS-Mg²⁺. As proposed by the reviewer and align with literature on PCAT1 (PMID: 26201595, 31934861), both these states are regarded as occluded. We have added this discussion to **Lines 497-498** and **1002-1003**, as well as in the working model section (**Lines 1534-1744**), and it is reflected in **Figure 8** with all PCATs structure resolved under specific functional state listed.

In short, the work is done nicely and there are certainly new advances here. However, the authors over-interpret some of their observations and do not do a service of accounting for prior studies done by others on similar systems. A revised version that addresses these concerns should be suitable for Nature Comm.

The manuscript has been thoroughly revised to remove the over-reaching statements and we included more discussions on the prior studies. Again, we would like to thank the reviewer again for the compliments on our work.

Reviewer #2 (Remarks to the Author):

The manuscript by Yu et al. explores the structural basis of competence-stimulating peptide (CSP) secretion by ComA, which plays a key role in bacterial quorum sensing. Authors reconstituted ComA in peptidiscs and demonstrated ComC cleavage was inhibited by the presence of ATP, similar to other PCAT family members. Using single particle cryo-EM, the authors solved the structure of ComA in several conditions: (1) ComA (E647Q)+ComC+ATP, (2) ComA+ComC, (3) ComA+ComC+ATPγS-Mg²⁺, (4) ComA+ATP-Zn²⁺. Authors identify a possible CSP binding pocket in their structure and note local rearrangements of the EC gate that seemed to depend on ATP-Mg²⁺ binding to the NBDs. Based on their structures, authors mutated key residues in the ComA cavity and show their effects on protease activity and a cell-based CSP secretion assay. The structure of ComA and identification of a potential CSP binding pocket are key findings that may potentially be of interest in the peptidase-containing ABC transporter field.

Thank you for the positive evaluation of our study.

However, we have a number of significant concerns about this work in its current form. While the cryo-EM structure of *Streptococcus pneumoniae* ComA is novel, the modeling of the cryoEM maps is not well executed and the interpretation of the structural data lacks rigor, particularly in areas where the authors make key claims in the manuscript such as the putative CSP binding site in ComA and the ATP-binding site in the NBD. What inferences can be drawn for the biochemical assays is also unclear to us, and while mutations are made that alter peptidase activity, the mechanistic basis is largely unexplored, even when the results are somewhat surprising. More broadly, the authors make strong claims throughout the manuscript despite having weak evidence supporting those claims. This manuscript will require substantial revision, which may change the key claims made.

We appreciate the constructive feedback from the reviewer. In reply, we have extensively revised the manuscript and refined the modelling of our cryo-EM maps. In addition, two new cryo-EM maps (C17A

mutant in the presence of ComC, and E647Q mutant in the presence of both ATP and Mg²⁺ bound), to elucidate the interplay between peptidase and ATPase activity and to substantiate the role of Mg²⁺ in the opening of the EC gate. Furthermore, we've provided more comprehensive explanations underpinning the rationale behind our biochemical studies. The strong claims have been adjusted or with more explanation/evidence for clarifications, to reflect a suitable degree of certainty as suggested. We hope that the reviewer will agree with us that the manuscript has been significantly improved with this round of revision.

MAJOR COMMENTS:

(1) In Lines 104-108: While a peptidisc may yield native-like activity for some membrane proteins, I think it cannot be assumed to be true always and must be demonstrated for each protein to be studied. Unlike a nanodisc, in a peptidisc there is no added lipid added around the membrane protein; it is effectively solubilized by the belt protein any residual detergent or lipids that aren't removed in the exchange. Many ABC transporters exhibit substrate-stimulated ATPase activity, but ComA does not in LMNG or peptidisc, which may indicate that it needs lipids to recapitulate its native mechanism (it is also possible that it just isn't simulated by substrate). We think the authors should be more conservative in their wording here. Also, Fig 1B seems to indicate a comparison between activity in "Detergent" vs "Nanodisc", but I think this is incorrect and that only peptidiscs were used here.

Agreed. In reply, we performed a control experiment in which ComA was reconstituted into Nanodiscs (**Figure S1A**), and we then assessed its ATPase activity both in the presence and absence of the substrate peptide. These results are now included in **Figure 1B**. Although still no substrate-stimulated ATPase activity was observed, we acknowledge that these in vitro reconstitution methods for ComA may not fully replicate the native lipid environment.

Thus, as suggested, the statement has been revised to avoid commenting on multiple approaches to solubilize membrane proteins, which is also outside of the scope of this study. Furthermore, we have cited a previous study on PCAT1 (**Lines 120-122**), where no evident substrate-stimulated ATPase activity was observed, aligning with our findings on ComA. We've also adopted a more cautious tone in our language (**Lines 117-122**). We appreciate your insightful comments.

(2) In Lines 139-147; Table S1: It seems like there are generally fewer crosslinked peptides detected in the ATP-bound sample versus apo. Might the crosslinks not observed in the ATP-bound sample be simply missed due to lower coverage? Can the data be analyzed in a more rigorous way? How many replicates were performed? No stats are reported indicating whether the differences are significant, and no indication of peptide abundance is presented.

We thank the reviewer for the comments on the cross-linking mass spectrometry data. We have added in more details in the text (**Lines 367-461**) indicating the number of replicates used, statistical cut-offs used for CLMS data analysis and number of unique cross-link sites identified between ATP-free, ATP-bound and across the two samples.

To directly demonstrate the dissociation of the PEP domain upon ATP binding, we conducted cross-linking mass spectrometry (CLMS) experiments on ComA in the presence and absence of ATP, cross-linked with disuccinimidyl sulfoxide (DSSO) with 2 biological replicates digested with trypsin and 2 biological replicates digested with chymotrypsin. Our CLMS analysis with 1% false discovery rate (FDR) cut-off identified a total of 46 unique cross-link sites across both states, with 27 cross-links found in ATP-free state only, 6 cross-links found in ATP-bound state only and 13 cross-links found in both ATP-free and ATP-bound states (**Figure S4, Table S1**). Cross-links which fall on the PEP region were mapped to the cryo-EM structure, with approximately 60% of cross-links within expected C α -C α distance of 30Å for DSSO cross-links.

The reviewer brings up a good point that the lower number of cross-links may be due to lower coverage of ATP-bound sample. We have added additional details that peptide counts across the samples are relatively similar which rules out the possibility of lower coverage.

Similar amounts of protein were analyzed for both ATP-free and ATP-bound states, with peptide counts of 385, 388 for ATP-free trypsin replicates; 493, 410 for ATP-bound trypsin replicates; 55, 56 for ATP-free chymotrypsin replicates and 28, 46 for ATP-bound chymotrypsin replicates. The lower number of cross-links observed between the PEP domain and ICgate in the ATP-bound state compared to the ATP-free state are likely to be due to the presence of ATP rather than overall protein amount.

(3) Figures often show error bars, but the legends don't state what they represent. They also don't state the number of replicates carried out, and whether the replicates were technical or biological.

We apologize for the oversight. The legends of Figures 1, 4, 6, 7, S6, S9 and S18 are revised.

(4) The authors model protein side chains and ligands in regions of their cryo-EM map that are lower resolution and contain weaker densities. This is problematic since the authors make claims in the manuscript related to these regions such as the CSP pocket and ATP-binding site, which may be an overinterpretation of their experimental maps. In several figures (eg. Fig. 3A, 4C, 4D, 4G), authors show cryo-EM densities carved around poorly modeled ligands which is deceptive since it masks the surrounding protein and makes the ligand appear to have stronger density.

For example, in Fig. 3A, the actual map doesn't look like this figure. The map contour would have to be unreasonably low and then carved to make a figure like this. This is misleading and how the map was carved or modified to make this figure should be stated clearly in the legend. In general, it is helpful to include information about contour level and how the density was generated (volume zoning radius, etc.) for any figure containing cryo-EM density.

Thank you for your valuable feedback, we have carefully addressed your suggestions in the revised version:

1. Regarding the EM map quality, we have improved the EM map quality of the Zn^{2+} bound structure, resulting in a better representation (**Figure 6C**). Additionally, we have resolved new EM structures to further support our points on the role of Mg^{2+} binding (**Figure 5C**).

2. To address concerns about clarity in our figures, we have ensured that all EM figures, including Fig 3A, 4C, 4D, and 4G, are now clearly labeled with contour levels and carving parameters.

Regarding the CSP-bound map, we acknowledge the potential for misleading interpretations due to the not-so-strong density. While considering the fact that the captured state represents the stage just before CSP release, and the identified pocket signifies the site for CSP release - we propose that the site may not tightly bind CSP molecules, facilitating subsequent release. Thus due to the highly intrinsic flexibility of the state/site needed for subsequent CSP release, the overall conformation at this state, as well as the bound CSP molecules, would not be of high resolution, as observed in our study.

To address this while acknowledging the challenge in improving the resolution, we have made four major adjustments: 1) We replaced the original figure with one without any carving (**Fig. 3A, left**), 2) We included a figure with EM density carved to highlight the putative bound CSP molecules observed at a map of medium/low quality, but with clear labeling of carving and contouring parameters (**Fig. 3A, right**), 3) In particular, we have added more comprehensive explanations supporting the captured density as CSP molecules and distinguishing it from other components (**Lines 700-710**), and 4) To maintain scientific rigor, we have significantly downplayed the claim and made it explicit that this is a putative CSP binding pocket with inherent ambiguity concerning the bound molecules and their binding orientation. Furthermore, in the final deposited model, we will refrain from assigning a specific identity to the bound molecule (**Lines 710-714**).

We hope these modifications enhance the quality and accuracy of the EM map representation and the overall presentation of our findings.

(5) In their CSP-bound-ATP-free map, two CSPs are modeled in the putative binding pocket in ComA. Since the densities for the two ligands are generally weak, a major question is "what is it?" and what would be an appropriate way to model it? For starters, this sample has probably seen a whole lot of other molecules along the way, including lipids and detergent. It would be very difficult for the authors know that this is the

peptide and not something else, and no other evidence presented in this work directly supports that the peptide binds this site. Second, even if this were density for the peptide (admittedly, it might be), in our opinion, this map does not allow the assignment of the direction of the peptide (N- vs C- termini) nor the sequence register. This is evident in the relatively poor map-model scores for chain C (0.5146) and chain D (0.5049) and relatively high clashcore (15) of the model. So at best, I think the authors could say they observe density at this site, which may be a peptide but potentially could be something else. In the deposited model, this ambiguity should be reflected by labeling anything explicitly modeled as UNK or UNL. Orthogonal evidence such as cross-linking mass spectrometry on a sample of ComA+ComC or binding assays between ComA+ComC could be useful in validating the interaction between CSP and ComA and whether this pocket is specific for this peptide.

Thank you for your valuable feedback, this concern has been roughly addressed from Point #4 as highlighted in purple. In short, we have taken the reviewer's suggestion seriously and made appropriate adjustments to downplay our claims regarding the binding site, emphasizing the inherent ambiguity surrounding the bound molecules (**Lines 700-714, and 727-728**).

In addition, we have provided more comprehensive explanations to support our proposal for CSP binding. First, throughout our study, we presented multiple EM snapshots, all purified using the same strategy. However, the captured putative CSP density was only strong in this specific sample (super weak in state 2 conformation), with the addition of substrate peptide. Thus this point is further cross-validated by other EM analyses conducted in our study.

Second, to validate the bound density as CSP molecules, we conducted a direct negative control study by removing ComC from the sample. This resulted in a structure with the EC gate closed and no CSP molecules captured (data not shown). So, this serves as another control experiment provides direct evidence supporting our propose of CSP as the molecule bound.

Moreover, the EM density exhibits a linear shape with multiple sidechain-like densities along it, which does not resemble lipids. Additionally, if the density represented something other than CSP, it would have likely been observed consistently in all our EM maps, trapping the enzyme in an inactive state. However, our EM maps exhibit different conformations, representing different functional states in the enzymatic cycle, which aligns with the behavior of the bound ligand, CSP.

Lastly, to further strengthen our model, we have conducted extensive functional tests, including in vitro (**Lines 727-969, Figure 4C, 4E and 4F**) and in vivo experiments (**Lines 1306-1498, Figure 7 and S18**), involving the mutation of all residues in this binding site. These functional data provide robust evidence supporting our proposed model as well.

So together, based on the evidence presented, we strongly suggest that the EM density corresponds to CSP molecules, and the site represents a putative CSP binding site. But without super high resolution, we also fully understand and agree with the reviewer's points and thus have incorporated all the suggestions into our revised manuscript.

(6) In their reconstruction of the CSP-bound-ATP-free map (Fig. S6), authors applied C2 symmetry whereas C1 is used for all other other reconstructions in this study. What is the authors' reasoning for applying symmetry? Previous structures of related transporters such as PCAT1 (PDB 6V9Z, Kieuvongngam et al., eLife, 2020) show that the peptide cargo binds asymmetrically to the transporter. Application of C2 symmetry may present artifacts in the reconstruction, particularly in the region where CSP binds. I would suggest that the authors try to improve the resolution of this region by performing 3D classification and focus refinement in C1. For example, 3D classification may reveal that only 1 CSP molecule is present and that it can occupy both halves of the binding pocket equally.

Great point. We have been very careful in our analysis, especially considering that PCAT1 is shown with only one peptide substrate bound. The EM data processing of this specific data set involved using C1 up to the step just before applying C2 symmetry, at which point 2 CSP molecules were visualized. As

mentioned in **Lines 697-698** and illustrated in the EM workflow of **Fig S8C**, we decided to apply C2 symmetry for the final refinement. A detailed comparison of the EM map with or without C2 symmetry is also included in **Fig S8G**.

In an attempt to confirm the number of bound molecules, we conducted focused classification with a mask at this region, it still clearly showed two but unfortunately, there was no significant improvement in resolution or map quality. Therefore, we did not include the results in the study. This lack of improvement may be attributed to two main factors:

- 1) The particle alignment in the 3D reconstruction already favoured the TMD region, which includes the CSP binding site, due to the NBD domain's higher flexibility and smaller size.
- 2) As mentioned in the previous response, this state represents a moment right before CSP release, where the bound substrate is expected to be in a highly flexible state to facilitate easy release for ABC transporters. Consequently, this intrinsic flexibility significantly limits the attainable resolution.

(7) In Lines 229-231: I would expect the opposite, that peptide bound to the TMD would inhibit PEP activity, but mutation of the putative binding site (presumed to reduce peptide binding, but not demonstrated) actually inhibits PEP activity. The authors don't provide a rationale for their observations. More broadly, the absence of ATPase stimulation by substrate complicates the authors' biochemical analysis, leading them to use peptidase activity as a readout instead. However, why substrate binding to the TMDs should affect peptidase activity is not clear, thus exactly what information the peptidase assay provides is not clear to us. The basis for this assay and the logic is unclear to me. Why is PEP activity a good readout for mutations in the putative peptide binding site? It seems like a binding assay would be more appropriate.

The rationale is now detailed in **Lines 728-735**. We agree the binding assay is a more direct solution, but we expect it to be technically challenging because the CSP-bound state appears to be transient. Thus, the biochemical assay is the method of choice. Thanks!

(8) In Lines 296-306: I would say this analysis is not well supported by the data. Just looking at Fig. 4, C-D, there is not great density for the surrounding side chains even, let alone a map that would allow one to determine whether Mg is bound or not. And while the authors are trying to interpret if individual atoms are present, one of the ATP- γ -S molecules built into the OF-open structure is misfit in the map such that the base is rotated ~ 180 degrees from how it is normally oriented in ABC transporters (Chain A, 801), though the map clearly indicates that this is incorrectly modeled. The other ATP- γ -S doesn't have this issue, but instead the sulfur of ATP- γ -S is only ~ 1.3 Å from the Mg, which is physically impossible. It would seem that the data is not sufficient for the authors to place even groups as large as adenosine accurately in these maps, so I question their ability to assess whether Mg is present or absent.

Thank you for your valuable comments. In response, we have made substantial revisions to related section (**Lines 1252-1304**). As part of our reply, we have conducted a direct comparative study by resolving a new EM map of the ComA (E647Q) mutant in the presence of both ATP and Mg²⁺ (**Lines 1252-1262**). This allows us to compare it with the previous map resolved in the presence of ATP only. Along with this, we have thoroughly refined and corrected the model, addressing all geometry and close clash issues for improved accuracy.

To ensure the reliability of our findings, we have excluded Q565 from the results due to the current resolution being insufficient for definitive conclusions regarding surrounding residues. However, with improved data processing, the Zn²⁺-bound map now exhibits clear density for H646, a key residue showing conformational changes in response to Mg²⁺ binding. Please refer to our revised **Fig 6** and the updated text at **Lines 1269-1276 and 1284-1286** for these findings. Additionally, we compared our results to the previously determined structure of the NBD domain alone in the presence of Mg²⁺, and both show a stretched conformation of H646, providing consistent results (**Lines 1272-1275, Fig S15**).

(9) Based on State 2 that was observed in their ComA+ComC+ATPyS-Mg²⁺ sample, authors claim that ATP-Mg²⁺ complex is needed to open the EC gate and release CSP. However, closer inspection of the cryoEM map of State 2 reveals that densities globally similar to those observed in the "CSP-bound"

structure are still present, perhaps only slightly weaker. Claims that this structure has released CSP are therefore not on solid ground.

Great observation, and thank you for pointing that out. We have now emphasized the weak density observed at the CSP-binding site in the text (**Lines 1010-1013**). We propose that this density arises from partial particles that still retain CSP molecules, which have not been fully released from the binding pocket.

As the cryoEM sample used for this analysis is the same as the one used for obtaining the CSP-bound structure, the only difference being the addition of ATP_{PrS}-Mg²⁺, we can infer that ComA initially contains CSP loaded into the EC-gate region. However, in the state 1 structure, representing the release of CSP upon the addition of ATP_{PrS}-Mg²⁺, the density at the CSP-binding site completely disappears. This suggests that in state 1, both bound CSP molecules are released.

In contrast, in state 2 where the EC-gate is open, weak density persists, indicating that some particles still contain CSP molecules that have not been fully released. In other words, we captured a conformation prior to the complete release of bound CSP molecules. We hope this clarifies the situation, and we appreciate your attention to detail. Thank you!

(10) In Lines 308-321: I think the rationale behind this experiment is not clear, nor are the biological implications. It is known that ABC transporters and many other ABC transporters use Mg, and occasionally can tolerate other divalents like Mn. It is known that Zn usually doesn't work well. Apart from demonstrating that this ABC transporter also doesn't use Zn very well, it's not clear to me what this adds beyond the OF-open structure in the presence of ATP- γ -S. Similarly, given Mg is abundant in the cell, under what circumstances would Mg not bind almost simultaneously with the ATP? This seems to be a major point of the paper, and comes back again in the discussion (In 387-393), but I think ATP and Mg may actually be pre-complexed in the cytoplasm and bind the transporter simultaneously. In the cell, there likely is never an ATP-only state, so the relevance of this seems very limited.

As suggested by the reviewer, we toned down the significance of the Mg²⁺ ion. Yet, we believe it is a good experiment to do because Zn²⁺ is a related divalent cation and is known to be toxic to pneumococcus, and been utilized by plants as a defense strategy toward pathogens (**Briefed in Lines 1288-1289**). Furthermore, the ATP-Zn²⁺ bound structure serves as a valuable mimic of the ATP-Mg²⁺ bound structure. This provides a useful tool for mechanistic interpretation by allowing a comparison with the nucleotide-bound structure in the absence of metal. Such a comparison validates that it is the ATP-Mg²⁺ complex, and not ATP alone, that triggers the opening of the EC gate, leading to CSP release. Also, these structures together provide a plausible mechanism underneath the opening of EC gate upon both ATP and Mg²⁺ binding (**Lines 1288-1304**).

(11) The resolution of the uploaded ATP-Zn ComA map seems much lower than what is shown/reported in Fig. S12 (3.8 Å-resolution). The uploaded map has very little side chain information, yet the authors have model all the residues of ComA into the map with an overall clashscore of 51. Is the reported resolution a typo? Sent us the wrong map as well as refined the model with sub-optimal parameters? Additionally, the authors have modeled in ATP-Zn into the map despite having poor resolution for the substrate as well as the surrounding protein. Since a lot of the protein is unaccounted, density that is zoned around the ATP-Zn in Figure 4G may also include unmodelled protein.

Apologies for the mistake in the previous map submission. We now have an improved map at around 3.9 Å resolution, displaying better quality with visible side chains at the ATP-binding site, including key residues like His676 in the NBD domain. The new map has been uploaded for your review, and we have incorporated a more careful comparison and discussion in **Lines 1278-1286** and **Fig 6C** to strengthen our analysis. Thank you for your understanding.

(12) In Lines 332-342: The interpretation of these experiments is confounded by the fact that the authors show in Fig. 3 that many of these mutants modulate PEP activity, and it seems like their impact of PEP activity generally mirrors their impact on "transport". What is being measured here seems to be a

combination of peptide cleavage and transport, yet it seems like the results could potentially be explained by the processing defect from Fig 3 with no change in transporter activity. Is transport actually affected at all by these mutants? Can the peptide be provided to the transporter in a way that uncouples the peptidase function from the transport function, perhaps by co-expressing soluble PEP domain and ComC, along with ComA with a deletion or mutation of the PEP domain?

As suggested by the reviewer, we perform complementation assays with the soluble PEP domains (**Fig. S18**). The results suggest the peptidase function couples to the transporter function, which agrees with the paradigm of the field. In other words, if the ComA mutants are defective in the peptidase, it will likely no longer be able to export CSP.

MINOR COMMENTS:

(1) In 29-30: "determine five cryo-EM structures in the presence or absence of CSP or ATP": Strictly speaking, this is incorrect, as 2 have ATP-gamma-S bound; similar issue In 89-90

Thank you for the suggestion. To mitigate any confusion, we have removed the number of structures resolved in this study to avoid confusion. Additionally, for the sake of precision, we've substituted the term "ATP" with "nucleotides".

(2) In 103: Perhaps "solubilized" would be a better word than "dissolved"?

Changed.

(3) Fig. S1D: Is there an error in the labels above the gel? The first lane after the ladder has no labels, but is perhaps ComC alone? But then the next lane, which is indicated to be ComC alone, appears to be fully processed ComC cleaved to CSP and leader. I am therefore not sure what to conclude from the other lanes.

The label for Fig. S1D has been revised for improved clarity, thanks!

(4) Figure S2. Would it be possible to color code the cross-links to better highlight the X-links that are unique to the ATP-free sample versus ATP bound sample.

We have colored coded Figure S2, **now the updated Figure S4**, to reflect cross-links in both ATP-free and ATP-bound sample as black, cross-links in ATP-free sample only as green, and cross-links in ATP-bound sample only in red.

(5) In 130-131: These results are interesting, but I am having trouble understanding how to interpret this result and the authors merely present the data. Isn't it a bit surprising that ATP vs non-hydrolyzable analogs have the same effect? I might have at least expected ATP+Mg to have an intermediate effect, since it can turn over while the other might be trapped in a particular conformation. Strictly speaking, the authors don't show the effect of Mg-alone or ATP-alone, so perhaps one can't rule out the possibility that Mg is inhibiting cleavage somehow? It is also unclear at this point if inhibition is mediated by ATP/analog binding to the NBDs of the ABC transporter, or somewhere else. A mutation that blocks ATP binding to the NBDs would provide evidence that inhibition is mediated via the NBDs.

Excellent points, thank you! As recommended, we conducted experiments to evaluate the effects of Mg²⁺ alone, ATP alone, and introduced a mutation that hinders ATP binding to the NBDs (D646N), in order to assess their respective impacts on peptidase activity.

In essence, regarding that the peptidase activity of the wild-type (WT) ComA enzyme was inhibited in the presence of ATP-Mg²⁺ compared to when the enzyme was exposed to ComC alone, we propose that this could be due to ATP hydrolysis being the rate-limiting step in ComA's function, while the peptidase-catalyzed reaction proceeds at a much quicker pace. Further details on the rational in and findings are elaborated in **Lines 328-440** and illustrated in **Figure 1F, S1D, S1E, and S1F**.

(6) In 160. 0.5 mM ATP written here but 2 mM ATP was used according to Methods (Line 495).

Corrected.

(7) In 185-192: The experiment and logic are not well described here. What is being mutated? How is making multiple mutations and destabilizing the protein contrary to the functional studies?

We have rewritten this section to improve clarity (**Lines 482-495**).

(8) In 194-195: Perhaps some comparison to past work would make sense here, as this was probably expected based on the structures of PCAT (PMID: 26201595), and perhaps also IrtAB (PMID: 32296173), rv1819c (PMID: 32296172), YbtPQ (PMID: 32076651), etc. Also, it seems like a structure of the ComA NBDs from another species was reported, but no comparison was made to this structure.

Thank you for your insightful suggestions. We have conducted volume calculation using 3V server for the central cavity of all recommended structures, including McjD, another peptide exporter with an available structure. Furthermore, a comparison has been drawn with the resolved NBD domain. These analyses have been incorporated in **Lines 475-477** (comparison to the resolved NBD structure), **Lines 497-508** (comparison study of central cavity to other ABC transporters), and in the **Figure S6A and S7**.

(9) In 203-206: This is confusing. If in the next section, evidence will be presented that CSP does not bind here, why speculate that it may bind here, but that it probably doesn't?

We have rewritten this section to improve clarity (**Lines 687-691**).

(10) Figure 3B: Color hydrogen bonds vs salt bridges different colors

Revised.

(11) Figure 3D: Colors of labels are switched. K labels should be blue

Corrected.

(12) In 269: No data for CSP release is shown. Section title should be reworded to not be so speculative.

Revised.

(13) In 297: I suggest deleting "high atomic resolution", as 3 Å is still very far from atomic resolution. This is clearly illustrated by my comment on In 296-306.

Deleted.

(14) It seems that the PEP domain is never visible in their maps, yet somehow the conformation of the transporter/nucleotide state is regulating the peptidase activity. How do the authors imagine this is happening?

We propose that the regulation of peptidase activity by nucleotide state arises from conformational changes. In brief, peptidase activity remains high when the PEP domain associates with the ABC transporter's primary structure, and the two NBD domains are separate. ATP binding, however, initiates PEP domain dissociation, resulting in inhibited peptidase activity. We substantiated this with peptidase assays involving various factors and mutants, a comparative EM study examining ComA trapped in ComC-bound and ATP-bound states, and a CL-Mass Spec study assessing samples with or without ATP. These results are detailed in a new section in the revised manuscript in **Lines 317-461**.

Our newly conducted comparative EM study provides most direct evidence (**Figure 2**). we used ComA C17A mutant (peptidase-inactive) in the presence of ComC, and E647Q mutant (ATPase-inactive) in the presence of ATP. The cryo-EM maps revealed a single conformation for each sample. The ComC-bound sample showed an approximate resolution of 6 Å, whereas the ATP-bound sample achieved a higher resolution of 3.1 Å.

As hypothesized, significant conformational changes were observed in the PEP and NBD domains under different conditions. In the presence of the peptide, two associated PEP domains were clearly visible. Meanwhile, the NBD domains were widely separated. In contrast, under ATP presence, the NBD domains were closely associated, and the PEP domains appeared to dissociate, suggesting their transition to a more flexible state.

(15) In 514-539. Steps for cryoEM image processing are sparse. It would be great if authors could include more details for readers about how they processed their data (e.g. criteria for excluding certain classes during 3D classification, when masks were applied, etc.)

Thank you for your suggestion. We have incorporated more details into the methods section to enhance its clarity (**Lines 2070-2085**).

Reviewers' Comments:

Reviewer #1:

Remarks to the Author:

The authors have done a stellar job of addressing concerns raised in the initial review, including additional experimental data. I strongly endorse this manuscript for publication in Nat. Comm.

Reviewer #3:

Remarks to the Author:

The manuscript by Yu et al. describes the structural characteristics of the quorum sensing-protein ComA and proposes a mechanism of ComA in processing and secretion of the competence-stimulating peptide CSP. The structures of ComA reported by the authors are novel and provide advanced molecular information to other known structures of PCAT family proteins. From the tracked changes provided, I can see that the authors have made substantial revision to improve the presentation and conclusiveness of the findings. The revised manuscript has now thoroughly discussed the limitation of the work raised by the reviewers and the authors have made appropriate adjustments to some of the strong claims and provided new structures and in vivo CSP export experiments to support their conclusion. I think the work is nicely designed and conducted and has important implications in understanding the quorum sensing system.

I have only a few concerns about the mechanism of ComA that the authors proposed at the end of the manuscript.

Canonical ABC transporters normally utilize energy from ATP binding or ATP hydrolysis through NBD to induce conformational changes of the substrate-bound TMDs to achieve cargo repulsion. Based on the mechanism that the authors claimed for ComA, it seems that the ATPase activity of ComA is redundant for its function. The cartoon working model in Fig 8 shows the closure of NBDs and IC gate as well as the conformational change of TMDs from IF to OF are all achieved by binding of CSP to the central cavity. CSP translocation/secretion is driven by the ionic interactions or charge repulsions from the lumen residues of the central cavity. The opening of the EC gate for CSP release is dependent on the binding of divalent ions and resetting of the closed EC gate is also induced by the release of Mg²⁺ (Fig8). It seems none of these events requires the ATPase activities of the transporter.

These claimed mechanisms can be confirmed in the future by the in vivo transport assay using mutants that directly affect ATPase (such as E647Q) or peptidase activities (such as C17A, with co-expression of CSN) to test if ATPase or peptidase are required for cargo transport. The current reduced CSP release caused by mutating CSP binding residues may be due to reduced substrate binding to the transporter.

Typos:

Line 55, "which."

Line 60, "structure.."

Line 219, "ad"

Line 388, "confomration"

REVIEWER COMMENTS

Reviewer #1 (Remarks to the Author):

The authors have done a stellar job of addressing concerns raised in the initial review, including additional experimental data. I strongly endorse this manuscript for publication in Nat. Comm.

We thank the reviewer for the highly positive feedback and enthusiastic recommendation of our revised manuscript for publication. Again, we deeply value the constructive comments from the previous review process, which significantly contributed to the improvement of our manuscript.

Reviewer #3 (Remarks to the Author):

The manuscript by Yu et al. describes the structural characteristics of the quorum sensing-protein ComA and proposes a mechanism of ComA in processing and secretion of the competence-stimulating peptide CSP. The structures of ComA reported by the authors are novel and provide advanced molecular information to other known structures of PCAT family proteins. From the tracked changes provided, I can see that the authors have made substantial revision to improve the presentation and conclusiveness of the findings. The revised manuscript has now thoroughly discussed the limitation of the work raised by the reviewers and the authors have made appropriate adjustments to some of the strong claims and provided new structures and in vivo CSP export experiments to support their conclusion. I think the work is nicely designed and conducted and has important implications in understanding the quorum sensing system.

We thank the reviewer for the thorough review, positive feedback, and support for the publication of our revised manuscript. It is really encouraging to us that the reviewer recognizes the significance of our structural insights into ComA and our proposed CSP processing mechanism in advancing our understanding of the quorum sensing system. Furthermore, we appreciate the acknowledgment of the revisions we made to enhance the overall quality of our manuscript.

I have only a few concerns about the mechanism of ComA that the authors proposed at the end of the manuscript.

Canonical ABC transporters normally utilize energy from ATP binding or ATP hydrolysis through NBD to induce conformational changes of the substrate-bound TMDs to achieve cargo repulsion. Based on the mechanism that the authors claimed for ComA, it seems that the ATPase activity of ComA is redundant for its function. The cartoon working model in Fig 8 shows the closure of NBDs and IC gate as well as the conformational change of TMDs from IF to OF are all achieved by binding of CSP to the central cavity. CSP translocation/secretion is driven by the ionic interactions or charge repulsions from the lumen residues of the central cavity. The opening of the EC gate for CSP release is dependent on the binding of divalent ions and **resetting of the closed EC gate is also induced by the release of Mg²⁺ (Fig8). It seems none of these events requires the ATPase activities of the transporter.**

These claimed mechanisms can be confirmed in the future by the in vivo transport assay using mutants that directly affect ATPase (such as E647Q) or peptidase activities (such as C17A, with co-expression of CSN) to test if ATPase or peptidase are required for cargo transport. The current reduced CSP release caused by mutating CSP binding residues may be due to reduced substrate binding to the transporter.

Thank you for your insightful comments. Indeed, our comparison of the structures – one with the OF-EC gate open in the presence of both ATPγS and Mg²⁺ and the other with the OF-EC gate closed in the presence of only ATPγS, but no Mg²⁺ – strongly suggests that the release of Mg²⁺ induces the transition of the EC gate from an open to a closed state, as depicted in Fig. 8. We propose that in the cellular context, to triggers the release of Mg²⁺, is driven by ATP hydrolysis. While the reason we captured this phenomenon in the presence of nonhydrolyzable ATPγS, might be attributed to the dynamic equilibrium between the binding and dissociation of Mg²⁺ in vitro; While in cell, the release of Mg²⁺ is accompanied with fast ATP hydrolysis.

We acknowledge that this aspect was not sufficiently clarified in our initial model, and we apologize for any confusion it may have caused. To rectify this issue, we have made specific clarifications to the model section at Line 898-902 in the marked file. In case line numbers vary when the file is opened on different computers, we have included the changes below, with the modified parts highlighted in yellow for clarity.

Before modification: "5) Finally, after the CSP molecules are released, the extracellular gate closes once more, resulted in an OF-occluded state as observed in PCAT1 as well ²⁹. This conformation may be facilitated by ATP hydrolysis, followed by ADP release from the NBD domain. ComA then reverts to the apo state, ready for the next cycle of CSP processing and secretion."

After modification: "5) Finally, after the CSP molecules are released, **Mg²⁺ dissociate from the ATP binding site**, the extracellular gate closes once more, resulted in an OF-occluded state as observed in PCAT1 as well ²⁹. This conformation **with Mg²⁺ dissociated** may be facilitated by ATP hydrolysis **in cell**, followed by ADP release from the NBD domain. ComA then reverts to the apo state, ready for the next cycle of CSP processing and secretion."

We are also grateful for your suggestion to validate our proposed model through future in-cell studies. Thank you for your valuable input and support.

Typos:

Line 55, "which."

Corrected with the wrong period sign removed, thank you!

Line 60, "structure.."

Corrected with an extra period sign removed.

Line 219, "ad"

Corrected to "and".

Line 388, "confomration"

Corrected to "conformation", thanks!